

# Seasonal variations in physical characteristics of aerosol particles at the King Sejong Station, Antarctic Peninsula

Jaeseok Kim[1], Young Jun Yoon[1,*], Yeontae Gim[1], Hyo Jin Kang[1,2], Jin Hee Choi[1], and Bang Yong Lee[1]

[1]Korea Polar Research Institute, 26 Songdomirae-ro, Yeonsu-gu, Incheon 21990, South Korea

[2]University of Science & Technology (UST), 217 Gajeong-ro, Yuseong-gu, Daejeon 34113, South Korea

*Correspondence to: Y.J. Yoon (yjyoon@kopri.re.kr)





**Abstract**

The seasonal variability of the physical characteristics of aerosol particles at the King Sejong Station in the Antarctic Peninsula was investigated over the period of March 2009 to February 2015. Clear seasonal cycles of the total particle concentrations (CN) were observed. The monthly mean $CN_{2.5}$

5  concentrations of particles with a particle size lager than 2.5 nm were the highest during the austral summer with a mean of $1080.39 \pm 595.05$ cm$^{-3}$ and were the lowest during the austral winter with corresponding values of $197.26 \pm 71.71$ cm$^{-3}$. A seasonal pattern of the cloud condensation nuclei (CCN) concentrations coincided with the CN concentrations, where the concentrations were minimum in the winter and maximum in the summer. We also estimated values of fit parameter $C$ and $k$ based on

10  measured CCN spectra. The values of $C$ varied from 6.35 cm$^{-3}$ to 837.24 cm$^{-3}$, with a mean of $171.48 \pm 62.00$ cm$^{-3}$. The values of $k$ ranged between 0.07 and 2.19, with a mean of $0.41 \pm 0.10$. In particular, the $k$ values during the austral summer were higher than those during the winter, indicating that aerosol particles are more sensitive to supersaturation ratio (SS) changes during the summer than they are during the winter. Furthermore, the effects of the origin and the pathway travelled by the air mass on

15  the physical characteristics of aerosol particles were determined. The modal diameter of aerosol particles that originated from the South Pacific Ocean showed seasonal variations; 0.023 μm in the winter and 0.034 μm in the summer for the Aitken mode and 0.086 μm in the winter and 0.109 μm in the summer for the accumulation mode.





## 1. Introduction

Aerosol particles in the atmosphere may be emitted directly from various natural and anthropogenic sources (i.e., primary aerosol particles) or produced by gas-to-particle conversion processes (i.e., secondary aerosol particles). They influence local and global climates directly by scattering and absorbing radiation and indirectly by acting as cloud condensation nuclei (CCN) or ice nuclei (IN) (IPCC 2013). The physical, chemical, and optical properties of atmospheric aerosol particles determine their impact on climate change. Although various studies on the effects of aerosol particles on climate change have been carried out, the direct and indirect climate effects are still unascertained (IPCC, 2013). Moreover, in order to understand the sources and the processes of the atmospheric aerosol particles, there should be a need to have their long-term observations at different regions because aerosol particles vary temporally and spatially.

The Antarctic region is highly sensitive to climate changes due to complex interconnected environmental systems (e.g. snow cover, land ice, sea-ice, and ocean circulation) (Chen et al., 2009). Previous studies show that the Antarctic Continent and the Antarctic Peninsula have experienced noticeable climate changes (Rignot et al., 2004; Steig et al., 2009; Pritchard et al., 2012; Schneider et al., 2012). The Antarctic Peninsula, in particular, has a warming rate of more than 5 times that of the other regions on earth (Vaughan et al., 2003; IPCC, 2013). The Antarctic climate system can be linked with aerosol particles by complex feedback processes that involve aerosol-cloud interactions. In addition, because there are less anthropogenic emission sources in Antarctica, it is a suitable place to study the formation and growth processes of the natural aerosol particles. For these reasons, the observation of the physical properties in Antarctica, i.e., total particle concentrations, size distributions and concentrations of black carbon and activated CCN, is necessary.

Over the years, measurements of aerosol particles have been widely conducted at various stations in Antarctica; notably: Aboa (Koponen et al., 2003; Virkkula et al., 2007; Kyrö et al., 2013), Amunsen-Scott (Arimoto et al., 2004; Park et al., 2004), Concordia (Järvinen et al., 2013), Halley (Rankin and



Wolff, 2003; Roscoe et al., 2015), Kohnen (Weller and Wagenbach, 2007; Hara et al., 2010), Maitri (Pant et al., 2011), Mawson (Gras, 1993), McMurdo (Hansen et al., 2001; Mazzera et al., 2001), Neumayer (Weller et al., 2015), Syowa (Ito, 1985; Hara et al., 2011b), and Troll (Fiebig et al., 2014). The Antarctic aerosol particles have been investigated with regard to their size distributions (Koponen

et al., 2003; Belosi et al., 2012), optical properties (Shaw, 1980; Tomasi et al., 2007; Weller and Lampert, 2008), chemical compositions (Virkkula et al., 2006; Weller and Wagenbach, 2007; Asmi et al., 2010; Hara et al., 2011a), and mass concentrations (Mazzera et al., 2001; Mishra et al., 2004). Some studies focused on aerosol transport in the upper atmosphere (Hara et al., 2011b) and new particle formation (Järvinen et al., 2013; Kyrö et al., 2013; Weller et al., 2015). Although various studies have

been performed, the measurements taken at the Antarctic Peninsula and the long-term observations of aerosol particles are still insufficient.

In this study, we continuously monitored the physical characteristics of aerosol particles at the Korean Antarctic station (King Sejong Station) in the Antarctic Peninsula from March 2009 to February 2015. Measurements for aerosol size distribution and concentrations of total aerosol number,

black carbon (BC), and CCN were carried out using various instruments. The main aim of this study was to determine the seasonal variations of the physical properties of aerosol particles in the Antarctic Peninsula. In addition, the physical characteristics of aerosol particles that originated from the ocean and continent of the Antarctic region were investigated with air mass back-trajectory analysis.

**2 Methods**

**2.1 Sampling site and instrumentation**

Continuous observations of the physical properties of aerosol particles have been carried out since March 2009 at the King Sejong Station (62.22°S, 58.78°W) in the Antarctic Peninsula. Detailed

information of the sampling site is given by Choi et al. (2008). In brief, the King Sejong Station is located on the Barton Peninsula of King George Island (KGI). The population density of KGI is higher than that of other regions in Antarctica due to the various research activities carried out from eight





permanent on-site stations. The observatory is located approximately 400 m southwest of the main buildings, which include the power generator and crematory of King Sejong Station. Thus, the northeastern direction (355°-55°) was designated as the local pollution sector because of the emissions from the power generator and crematory at the station. We therefore discarded data from the local

pollution sector to improve data quality, and data where BC concentrations were higher than 100 ng m$^{-3}$ were also discarded. In this study, we present the analysed results of the physical characteristics of aerosol particles obtained during March 2009 to February 2015.

The physical characteristics of aerosol particles were continuously observed with various instruments that included two condensation particle counters (CPCs), an aethalometer, a cloud

condensation nuclei counter (CCNC), and a scanning mobility particle sizer (SMPS). The observation methods are shown in Fig. 1.

We used a laminar flow air sampling system for the sampling of aerosol particles. An aerosol inlet with a 10 cm diameter without a wind sector controller was placed on the roof of the observatory (Fig. 1). The total flow rate of the sample air was maintained at 150 lpm.

Total particle number concentrations were examined with two CPCs: a TSI model 3776 that measured particles > 2.5 nm in diameter and a TSI model 3772 that measured particles > 10 nm. Sample aerosol flow rates of CPC 3776 and CPC 3772 were 1.5 lpm and 1.0 lpm, respectively.

The aethalometer was used to measure the concentration of light absorbing particles at two wavelengths (370 and 880 nm). In this study, we used the results obtained by measuring light

absorption at 880 nm to determine the BC concentrations. The flow rate of the sample was constant at 5.0 lpm. The main purposes of measuring BC concentrations were to investigate long-range transport aerosol particles and to assess the influence exerted by local pollution.

To measure the CCN concentrations, a CCNC (DMT CCN-100) was used at five different supersaturation ratios (SS) (0.2, 0.4, 0.6, 0.8, and 1.0 %) and total flow rate of 0.5 lpm. The CCN

concentrations were decided by exposing aerosol particles at supersaturated conditions and then

counting only the number of only activated droplets with a detector. The sampling duration was set at approximately 5 min for each SS value (except the 0.2 % SS) before it was changed to the next SS value. For a 0.2 % SS, CCN concentrations were measured for 10 min because it required additional time to achieve stability after completing measurements at a 1 % SS.

Aerosol size distributions were continuously measured with the SMPS, which consisted of a differential mobility analyser (DMA), a CPC (TSI 3772), a control unit, an aerosol neutralizer (soft x-ray), and a data logging system. The resolution of scanning time was set to 120 s for mobility particle diameters from 0.01 to 0.30 µm. A closed sheath-air loop with a diaphragm pump in the control unit was used to maintain the sheath flow of DMA. The flow rate of sheath air of DMA was 10 lpm. The
ratio of aerosol flow to sheath flow of DMA was 1:10.

Besides, meteorological parameters including temperature, relative humidity (RH), wind speed (WS), wind direction (WD), pressure, and UV and solar radiation were also continuously monitored over the entire observation period.

**2.2 Back-trajectory analysis**

In order to associate the physical properties of aerosol particles to their source areas for the sampling periods, the air mass back trajectory analysis were conducted using the Hybrid Single-Particle Lagrangian Integrated Trajectory (HYSPLIT) model (Stein et al., 2015) (http://www.arl.noaa.gov/HYSPLIT.php). For every 6 h period, 120-h air mass back trajectories were
analysed, ending at heights of 100m, 500m, and 1500m above the ground level of the sampling site. The results where the origin and pathway of the air masses for at least 12 h were similar at three different heights were used for the analysis in this study. Based on this analysis, we have classified the air mass into four groups according to their origin and pathway: two continental regions (South America and Antarctica) and two oceanic areas (South Atlantic and South Pacific Ocean), as are shown
in Fig. 2.



## 3 Results and Discussion

### 3.1 Meteorological conditions

Fig. 3 depicts monthly variations of the meteorological parameters measured from and automatic weather system (AWS) during the whole observation period. The temperature varied between −19.5 °C and +5.8 °C, with a mean of −2.4 ± 2.1 ºC and the RH was between 60 % and 100 %, with a mean of 87.9 ± 3.3%. As mentioned previous studies (Kwon and Lee, 2002; Mishra et al., 2004), the observation site was relatively humid and warm condition compared to other Antarctic stations due to the effect of a marine environment. Values of the solar radiation varied between 2.3 W m$^{-2}$ and 375.4 W m$^{-2}$, with a mean of 81.2 ± 38.9 W m$^{-2}$.

### 3.2 Seasonality in the physical characteristics of aerosol particles

#### 3.2.1 Total particle number concentrations

Fig. 4 shows the monthly mean particle condensations (CN) measured with two types of instruments (TSI CPC 3776 and 3772) over the period from March 2009 to February 2015. All the seasons mentioned in this study are austral seasons. As can be seen in Fig. 4, there is an evident seasonal cycle of CN concentrations, which are the maximum in the summer (DJF) and minimum in the winter (JJA). The maximum concentrations of particles larger than 2.5 nm ($CN_{2.5}$) and larger than 10 nm ($CN_{10}$) were approximately 2000 cm$^{-3}$ in December 2012 and about 800 cm$^{-3}$ in December 2009, respectively. The minimum values of $CN_{2.5}$ and $CN_{10}$ concentrations were approximately 110 cm$^{-3}$ and 90 cm$^{-3}$ in August 2013, respectively. There are no significant anthropogenic sources of aerosol particles in Antarctica, therefore, our results were in good agreement with the results of previous studies from other Antarctic stations (Jaenicke et al., 1992; Gras, 1993; Virkkula et al., 2009; Weller et al., 2011). For instance, Virkkula et al. (2009) reported long-term daily average CN concentrations over the period from November 2003 to January 2007 from observations at Aboa, the Finnish Antarctic research station at a coastal region in Antarctica. The maximum monthly average CN concentrations were observed in February and the minimum concentrations were measured in July, which is the darkest



period of the year. The cause of the clear seasonal cycle of CN concentrations may be attributed to the formation process of aerosol particles. The major compounds of aerosol particles found at coastal Antarctic regions were non-sea-salt sulphate and methanesulphonate (MSA) derived from oxidation of dimethyl sulphide (DMS) produced by phytoplankton (Weller et al., 2011). The DMS concentrations increase sharply when biological activity is enhanced due to increasing temperatures and solar radiation (Virkkula et al., 2009). Since our sampling site was in the Antarctic Peninsula, ocean biological activity was considered to be an important factor in the particle formation and growth of aerosol particles. The CN concentrations typically increase in the summer due to high biological activity, while they decrease in the winter when biological activity is low. To better understand the effect of temperature and solar radiation intensity on $CN_{2.5}$ concentrations, we compared the relationship between monthly mean $CN_{2.5}$ concentrations and solar radiation intensity, and monthly mean $CN_{2.5}$ concentrations and temperature. The correlation coefficient between $CN_{2.5}$ and the solar radiation intensity (opened circle; $R^2=0.621$) was higher than that between $CN_{2.5}$ and temperature (opened triangle; $R^2=0.419$), as shown in Fig. 5. Our results suggest that the $CN_{2.5}$ concentrations may be more closely coupled with solar radiation intensity than with temperature.

A more detailed comparison of the monthly trends in the $CN_{2.5}$ and $CN_{10}$ concentrations is presented in Fig. 6. The monthly mean CN concentrations increased from September to February mainly during the austral spring and summer periods. The CN concentrations sharply decreased from March and remained stable from April to August. In particular, the $CN_{2.5}$ concentrations during the summer period increased sharply compared to the $CN_{10}$ concentrations, the increase was probably due to new particle formation. High solar radiation and temperature and low RH values during the summer are conducive to the new particle formation (Hamed et al., 2007).

### 3.2.2 Cloud condensation nuclei (CCN) concentrations

Fig. 7(a) shows the monthly mean CCN concentrations at the SS value of 0.4 % over the period from March 2009 to February 2015. There is a long gap in data from July 2011 to December 2013





because data were not collected due to a faulty CCN counter. We found monthly variations in the CCN concentrations with the maximum values being observed during the summer periods (DJF) and the minimum concentrations were observed during the winter periods (JJA). The monthly mean CCN concentrations were in the range of 20.63 cm$^{-3}$ in July 2009 and 227.52 cm$^{-3}$ in January 2014, with a

mean of 112.80 ± 39.05 cm$^{-3}$. It was similar to the seasonal cycle of the CN concentrations. Fig. 7(b) also shows seasonality in CCN concentrations at an SS value of 0.4 %. The CCN concentrations gradually decreased from February and remained stable during the winter, while the CCN concentrations from September increased sharply, as is shown in Fig. 7(b). The maximum CCN concentration in January was 199.89 ± 37.07 cm$^{-3}$ and the minimum CCN concentration in August was

42.13 ± 14.51 cm$^{-3}$. This clear seasonality of CCN concentrations is probably caused by the seasonal trend of CN concentrations. As shown in Fig. 6, CN$_{10}$ concentrations as well as CN$_{2.5}$ concentrations increased during the summer. In addition, the aerosol size distributions measured by SMPS showed that concentrations of accumulation mode particles in the range of 100 and 300 nm as well as Aitken mode particles during the summer increased significantly, as can be seen in Fig. 8. Accumulation mode

particles can easily act as CCN compared to nuclei or Aitken mode particles (Dusek et al., 2006), hence CCN concentrations increase during the summer and decrease during the winter.

In order to indirectly investigate the chemical characteristics of aerosol particles activated to CCN, the CCN spectra were examined in greater detail. An analysis of the cumulative CCN concentrations shown as a fraction of the CCN concentration measured at the SS of 1.0 % was carried out, and the

results are shown in Fig. 9. Here, fractions of the CCN concentrations were estimated by dividing the CCN concentrations at each SS value by the total CCN concentrations at the SS of 1.0 %. Although a clear seasonal trend of CCN concentrations with a maximum during the summer and a minimum during the winter was presented, as mentioned earlier, the fraction of CCN concentrations at the SS value of 0.2 % in activated CCN concentrations showed a different pattern with a maximum value in

July and a minimum value in December, as shown in Fig. 9. The numbers at the top of Fig. 9 represent





mean CCN concentrations at the SS values of 1.0 %. The fraction of particles activated to CCN at the

SS value of 0.2 % during the summer and the winter was $0.49 \pm 0.07$ and $0.62 \pm 0.06$, respectively.

The fraction at the SS value of 0.2 % during the winter (JJA) was similar to those measured in Mace

Head and Finokalia, which are regions representative of a marine environment (Paramonov et al.,

2015). Our observations suggest that the major components in the aerosol particles that are activated

to CCN at an SS of 0.2 % should be hygroscopic sea salts during the winter, while compounds less

hygroscopic than sea salt would be dominant during the summer.

    Fig. 10 illustrates the seasonal variations in the mean activation ratio of CCN concentrations at an

SS of 0.4 % to the CN concentrations measured from two CPCs (TSI 3776 and 3772). The mean values

of activation ratios of $CCN/CN_{2.5}$ and $CCN/CN_{10}$ were about $0.33 \pm 0.10$ and $0.40 \pm 0.08$, respectively.

Our results suggest that hygroscopic compounds were less dominant in the aerosol particles at our

sampling site compared to levels in aerosol particles in the Artic regions (Lathem et al., 2013).

Although clear changes were observed in the monthly variation in the CN and CCN concentrations as

shown in Fig. 6 and Fig. 7(b), it was seen that the activation ratio ($CCN/CN_{10}$) was similar regardless

of seasonality. The reason that no clear change is observed in the activation ratios at the King Sejong

Station in the Antarctic Peninsula, might be the variation of the concentrations of accumulation mode

particles, as can be seen in Fig. 8. The lower activation ratios in September and November are mainly

because of the size and chemical properties of aerosol particles. Both, the size and chemical

components of aerosol particles may have a large impact on the activation ratio (Dusek et al., 2006;

Leena et al., 2016). The concentrations of Aitken mode aerosol particles increased sharply compared

to their concentrations in August. Thus, the activation ratio decreased dramatically. Unfortunately, we

did not confirm aerosol size distribution because our aerosol size distribution data in November was

unreliable due to malfunctioning instruments.

    The CCN concentrations at SS values can be represented by a power-law function, defined by

Twomey (1959):





$$N_{CCN} = C \cdot (SS)^k \qquad (1)$$

where $N_{CCN}$ is the concentration of CCN at given a supersaturation values (SS), $C$ and $k$ are coefficient

constants estimated from CCN spectra. The values of $C$ varied from 6.35 cm$^{-3}$ to 837.24 cm$^{-3}$, with a

mean of $171.48 \pm 62.00$ cm$^{-3}$. The values of $k$ range between 0.07 and 2.19, with a mean of $0.41 \pm 0.10$.

The monthly variations of $k$ and $C$ values are also summarized in Fig. 11. A comparison with CCN

concentrations indicated that the values of $k$ during the austral winter (June) were also the lowest (0.29

$\pm 0.06$), while during the summer (December) they were the highest ($0.55 \pm 0.13$). Based on this result,

aerosol particles activated to CCN during the summer are expected to be more sensitive to SS changes

than those during the winter.

### 3.2.3 Black carbon (BC) concentrations

Fig. 12 shows variations of monthly mean BC concentrations over the whole sampling periods. The

BC concentrations varied between 7.18 ng m$^{-3}$ and 140.30 ng m$^{-3}$, with a mean of $64.68 \pm 12.17$ ng m$^{-3}$. The BC concentrations observed at our station were slightly higher than those at other stations in

Antarctica (Bodhaine, 1995; Wolff and Cachier, 1998; Pereira et al., 2006; Weller et al., 2013). For

instance, the annual mean BC concentrations at the South Pole, Halley, Neumayer, and Ferraz station

were 0.65, 1.0, 2.6, and 8.3 ng m$^{-3}$, respectively. Additionally, no clear seasonal patterns were observed

in our study throughout the entire observation period. However, clear seasonal patterns in previous

studies were observed at other stations in Antarctica (Wolff and Cachier, 1998; Weller et al., 2013).

Wolff and Cachier (1998) showed seasonal cycles of BC measured at the Hally station and South Pole

with a Aethalometer. They found that although BC concentrations varied depending on the sampling

site, the BC concentrations decreased during the austral winter (JJA) and increased during the austral

summer (DJF). Contrarily, according to Pereira et al. (2006), although BC concentration during the

summer increased slightly, no clear seasonal trends were observed unlike the results measured by Wolff



and Cachier (1998). This suggests that the BC concentrations are dependent on the sampling site and the long-range transport of air masses.

### 3.3 Effect of air mass trajectory on the physical properties of aerosol particles

5       In this section, the effect of the origin and pathway of air mass on the physical characteristic of aerosol particles is presented. As mentioned earlier in Sec. 2.3, we classified air masses into four groups based on air mass back trajectory analysis. The wind data and aerosol characteristics with the four types of air masses during the entire observation period are listed in Table 1. Although they are unreliable due to the low observation frequency compared to the other types of air masses, as shown

in Table 1, the BC and CCN concentrations were the highest when the air mass originated from the continent of South America (Case 1). This might be due to anthropogenic influences at the source and the aging of aerosol particles. The $CN_{10}$ concentrations were similar regardless of the origin and pathway of air masses, whereas an enhancement of the $CN_{2.5}$ concentrations was observed when the air mass originated from the ocean (Case 2 and 4). This is probably due to the high biological activity

in the South Atlantic and South Pacific Oceans during the summer (DJF) period. A more detailed comparison, excluding the results of Case 1 of the CN concentrations based on the air mass analysis is shown in Fig. 13. Minimum concentrations of aerosol particles ($CN_{2.5}$ and $CN_{10}$) originating from the ocean (Case 2 and 4) were observed from April to September, whereas concentrations of aerosol particles ($CN_{2.5}$) originating from the South Atlantic (Case 2) and the South Pacific (Case 4) Oceans

were the highest in November and February, respectively. Here we found that the peak month of the $CN_{2.5}$ concentrations had discrepancies in accordance with the air mass history. This is probably due to difference in chemical compounds that contributed to aerosol formation processes and/or in variations of biogenic activity according to the origin and transport pathway of air masses. To verify this, further studies on chemical compositions of aerosol particles need to be carried out in the future.

When air masses were transported from the South Pacific Ocean to the King Sejong Station (Case 4), the seasonality of aerosol size distribution was also investigated. The lognormal fitted aerosol size



distribution ranged from 0.01 to 0.3 μm is presented in Fig. 14. The modal diameters with standard deviation and number concentrations are summarized in Table 2. It is obvious that the modal diameters during the summer are larger than those during the winter for, both Aitken and accumulation modes: 0.023 μm in the winter and 0.034 μm in the summer for the Aitken mode and 0.086 μm in the winter

and 0.109 μm in the summer for the accumulation mode. The number concentrations for the summer are also higher than the value for the winter for the Aitken and accumulation modes, $49.16 \pm 3.88$ cm$^{-3}$ during the winter and $304.36 \pm 20.10$ cm$^{-3}$ during the summer for the Aitken mode and $44.78 \pm 14.24$ cm$^{-3}$ in the winter and $140.25 \pm 10.64$ cm$^{-3}$ in the summer for the accumulation mode. The enhancement of number concentrations for the Aitken mode during the summer should be linked to new particle

formation over oceans as a product of biological activity. The spring and autumn seasons show intermediate values. Our results are similar those of previous laboratory and field experiments (Sellegri et al., 2006; Yoon et al., 2007). O'Dowd et al. (2004) suggested that primary formation processes play a significant role in marine aerosol production in the North Atlantic Ocean. In addition, the contribution of biological organic compounds to the marine aerosol distribution might be dominant

(Kim et al., 2015).

**4 Summary and conclusions**

The seasonal variations in the physical characteristics of aerosol particles at the King Sejong Station (62.22°S, 58.78°W) in the Antarctic Peninsula were investigated based on the in-situ measured aerosol

data for the period from March 2009 to February 2015. An obvious seasonal variation of particle number concentrations (CN) exists, with the maximum concentrations in the austral summer (DJF) and the minimum concentrations in the winter (JJA). The maximum CN concentrations of particles larger than 2.5 nm ($CN_{2.5}$) and 10 nm ($CN_{10}$) were approximately 2000 cm$^{-3}$ in December 2012 and about 800 cm$^{-3}$ in December 2009 and February 2015, respectively. In particular, $CN_{2.5}$ concentrations

increased sharply during the summer compared to $CN_{10}$ concentrations, suggesting that the particle formation processes were probably driven by the high biological activity during the season.



In addition, we presented the seasonality of CCN. The maximum mean CCN concentration of $199.89 \pm 37.07$ cm$^{-3}$ was measured in January and the minimum mean CCN concentration was $42.13 \pm 14.51$ cm$^{-3}$ in August. The activation ratio (CCN/CN$_{10}$) of aerosol particles at the King Sejong Station $(0.40 \pm 0.08)$ in the Antarctic Peninsula was lower than those at the Arctic sites (0.52), indicating that less hygroscopic compounds in aerosol particles should be dominant. We also estimated $C$ and $k$ values from measured CCN results at certain SS values. The values of $C$ varied between 6.35 cm$^{-3}$ and 837.24 cm$^{-3}$, with a mean of $171.48 \pm 62.00$ cm$^{-3}$. The values of $k$ ranged between 0.07 and 2.19, with a mean of $0.41 \pm 0.10$. The $k$ values during austral the summer periods (DJF) were higher than those during the winter periods (JJA).

Based on the backward trajectory analysis, we classified the air mass into four groups according to their origin and pathway: two continental regions (South America and Antarctica) and two oceanic areas (South Atlantic and South Pacific Ocean). We found that most air masses originated from the oceanic areas. Although the BC and CCN concentrations were the highest when the air mass originated from the South American continent, the results are not significant because only a small amount of data was analyzed. The CN$_{10}$ concentrations were analogous regardless of origin, whereas CN$_{2.5}$ concentrations showed differing values. The CN$_{2.5}$ concentrations that originated from oceanic areas (Case 2 and 4) were higher than those from continental regions (Case 3), in particular, the CN$_{2.5}$ concentrations show clear seasonal variations; minimum concentrations from April to September and maximum concentrations in November from the South Atlantic Ocean (Case 2) and in February from the South Pacific Ocean (Case 4). Furthermore, in terms of Case 4, an analysis of aerosol size distributions in the 0.01-0.3 μm range was performed. The modal diameters also showed seasonal variations, 0.023 μm in the winter and 0.034 μm in the summer for the Aitken mode and 0.086 μm in the winter and 0.109 μm in the summer for the accumulation mode.

Overall, this study is the first of its kind to analyze seasonal variations in the physical characteristics of aerosol particles in the Antarctic Peninsula. The aerosol particle formation process is still not fully





understood, and thus, more studies should be necessary to determine seasonal variations in the chemical characteristics of atmospheric aerosols.

## Acknowledgements

We would like to thank the many technicians and scientists of the overwintering crews. This work was supported by a Korea Grant from the Korean Government (MSIP) (NRF-2016M1A5A1901769) (KOPRI-PN16081) and the KOPRI project (PE16010).

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





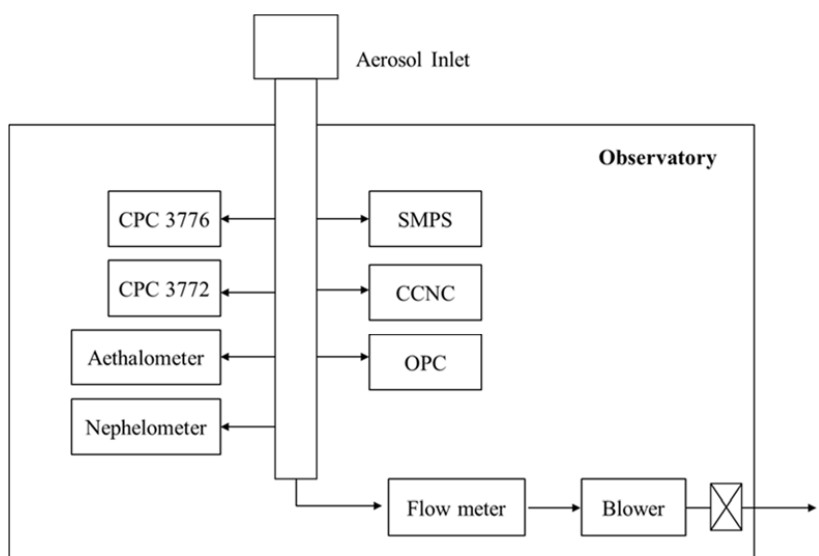

CPC: condensation particle counter
SMPS: scanning mobility particle sizer
CCNC: cloud condensation nuclei counter
OPC: optical particle counter

5    Figure 1. A schematic diagram for the observation methods used in this study.



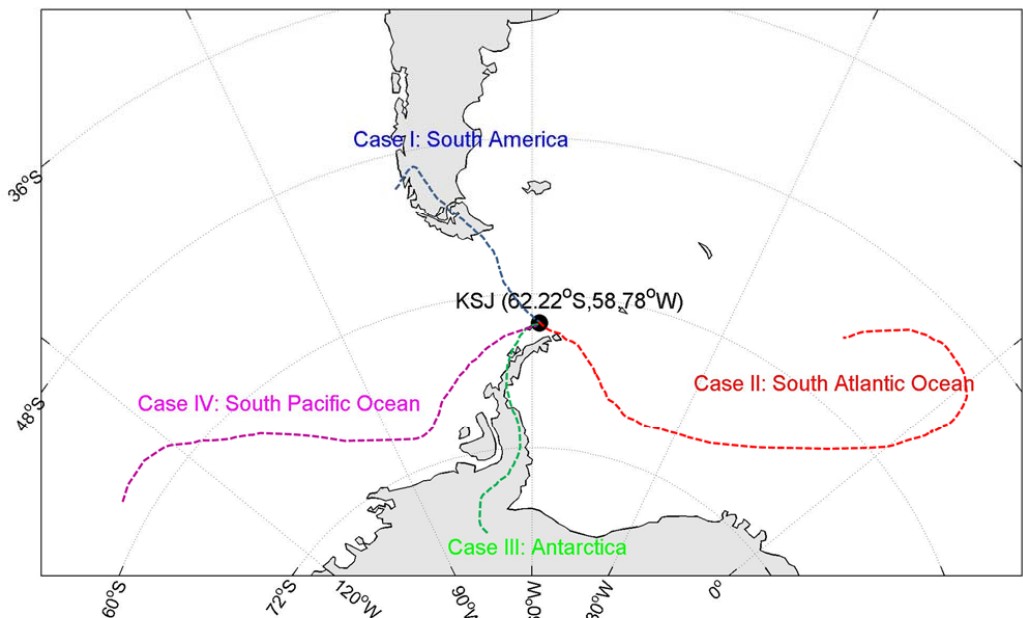

Figure 2. Map of the sampling site (62.22°S, 58.78°W; black circle) and classification of the four cases
according to the origin and pathway of the air masses. Dot lines represent example of back trajectories
according to cases.



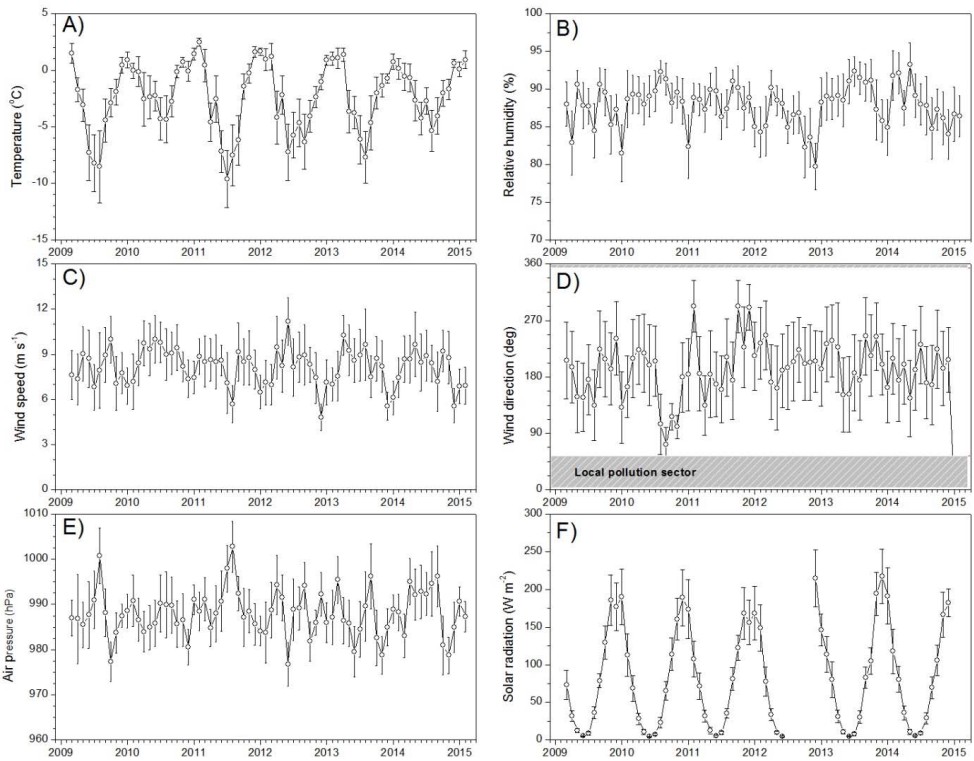

Figure 3. Monthly mean variation of (a) temperature, (b) relative humidity, (c) wind speed, (d) wind direction, (e) air pressure, and (f) solar radiation over the period from March 2009 to February 2015. The shaded area in Figure 3(d) represents the wind direction for the local pollution sector.





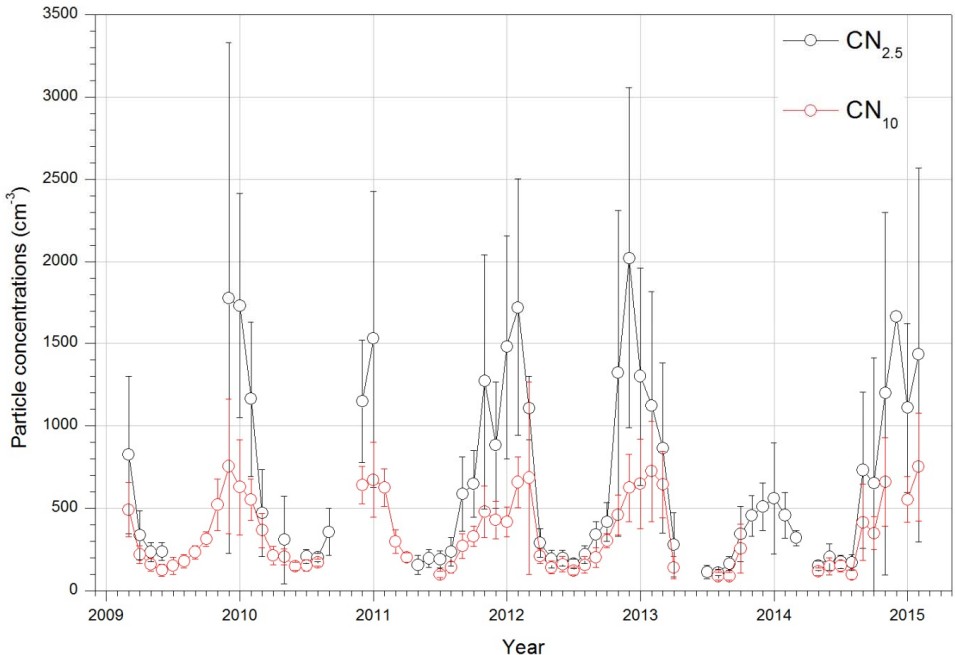

Figure 4. Monthly variations of mean $CN_{2.5}$ (black opened circle) and $CN_{10}$ (red opened circle)
5    concentrations with a standard deviation from March 2009 to February 2015.

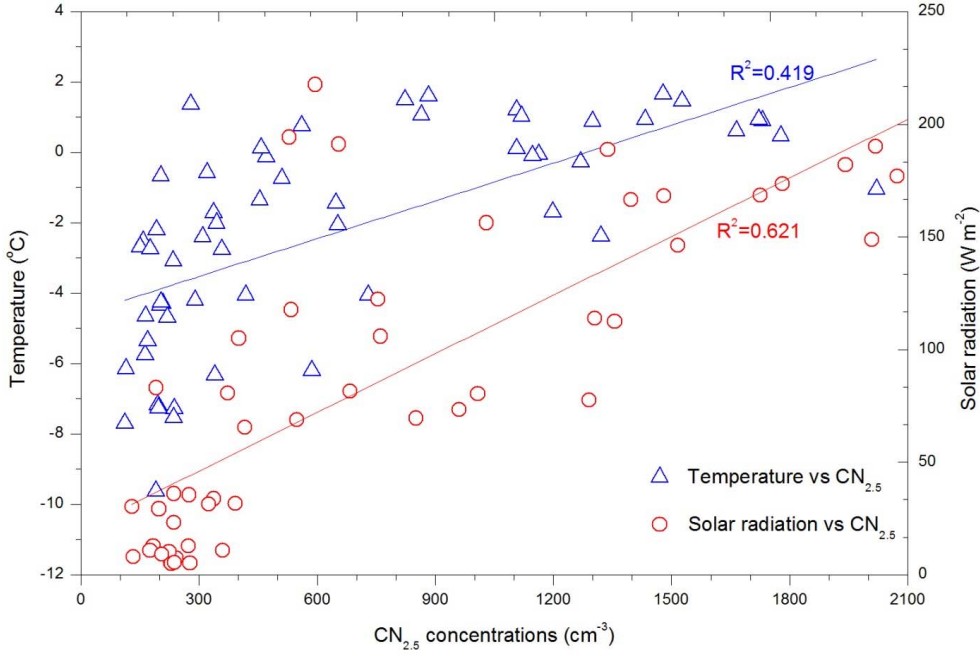

Figure 5. Scatterplot diagram of monthly mean CN$_{2.5}$ concentrations and monthly mean temperature (blue opened triangle) or monthly mean solar radiation intensity (red opened circle). Blue and red solid lines are a regression lines.




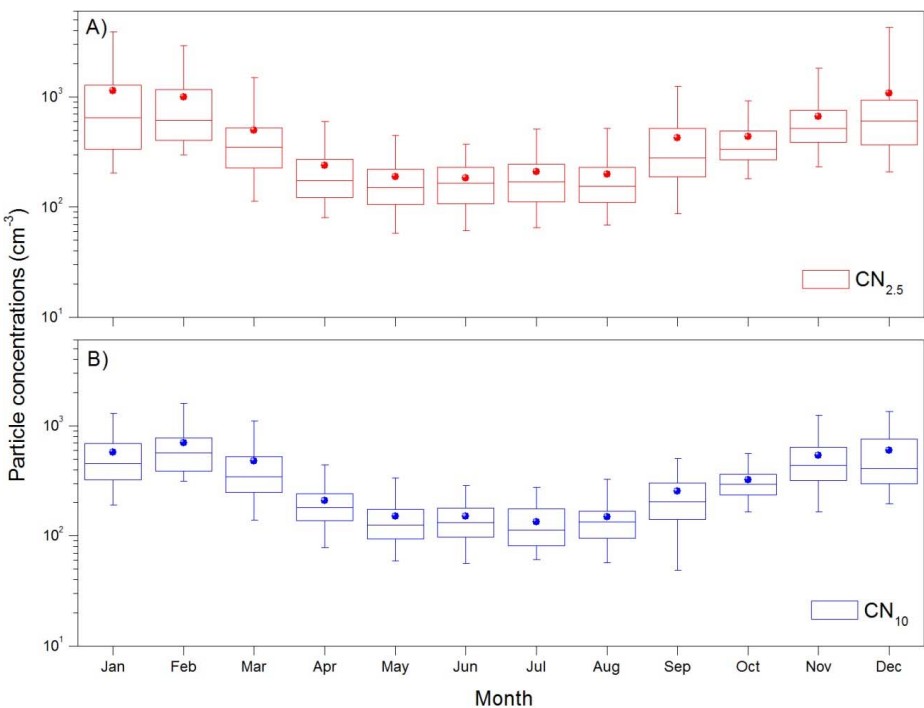

Figure 6. Box plots of seasonality of (a) $CN_{2.5}$ and (b) $CN_{10}$ concentrations. Lines in the middle of the boxes indicate sample medians (mean: circle), lower and upper lines of the boxes are the 25th and 75th percentiles, and whiskers indicate the 5th and 95th percentiles.



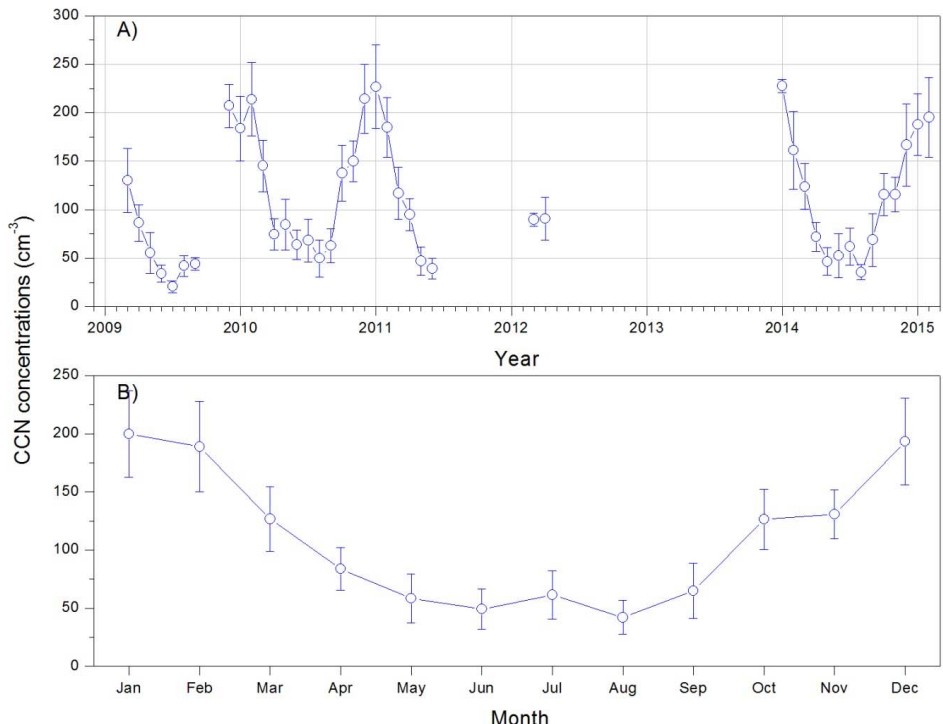

Figure 7. (a) Monthly mean CCN concentrations at the SS of 0.4 % with a standard deviation from March 2009 to February 2015 (b) Seasonal variation of mean CCN concentrations at the SS of 0.4 % with a standard deviation.



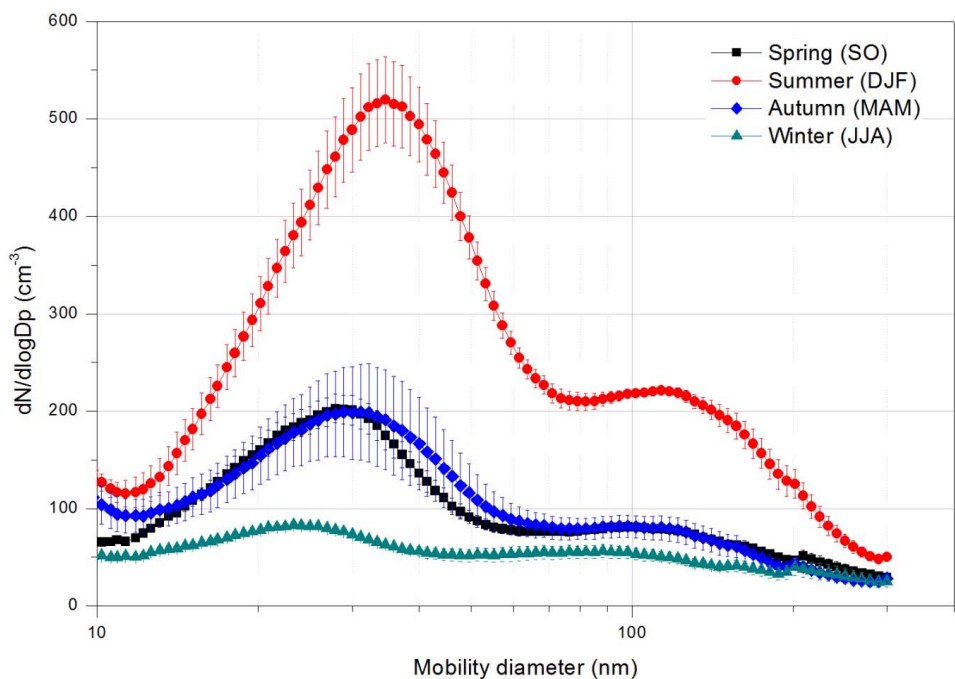

Figure 8. Seasonal mean aerosol size distribution measured by the SMPS at the King Sejong research station over the period from March 2009 to February 2015.





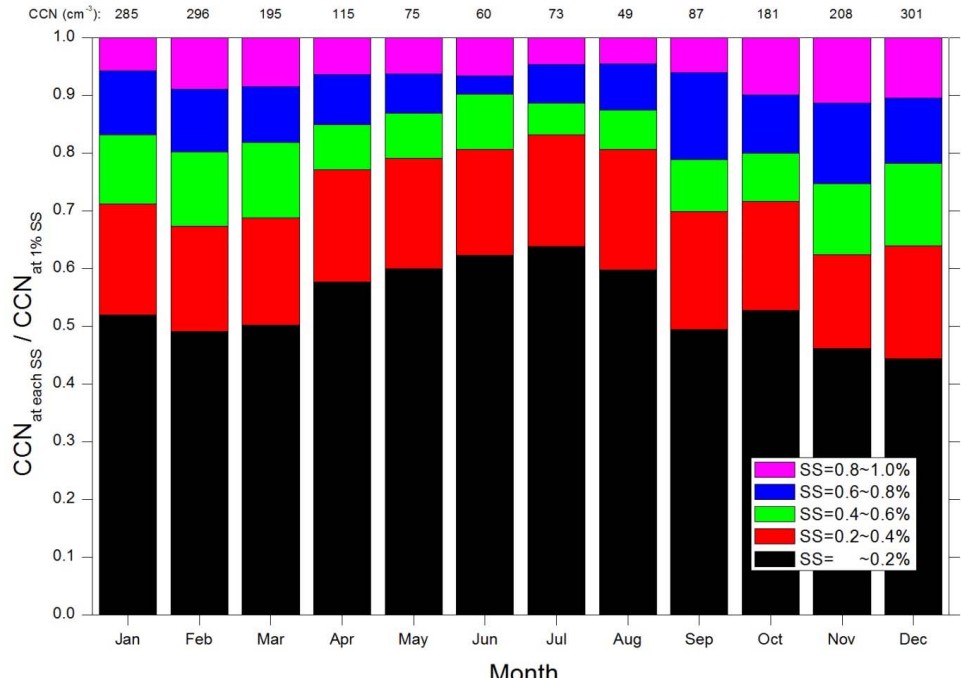

Figure 9. Monthly mean cumulative CCN concentrations shown as fractions of the CCN concentration at the SS of 1.0 %. Colours indicate the SS bins. The number at top of figure represents monthly mean CCN concentrations at the SS values of 1.0 %.




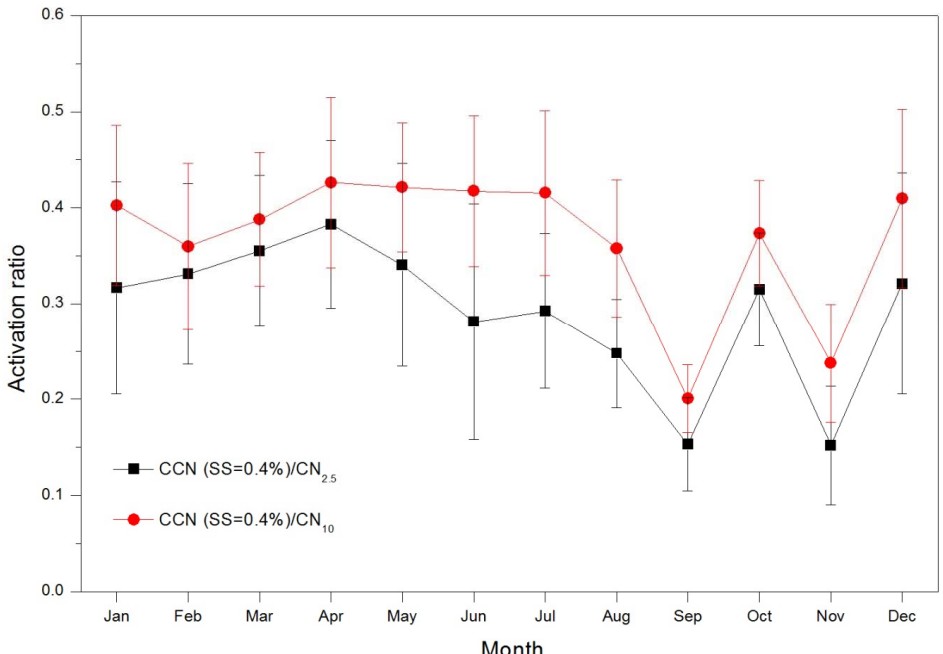

Figure 10. Comparison of the seasonal mean variation of the activation ratio between measurements
5    (CPC 3776 and CPC 3772) by two CPCs.



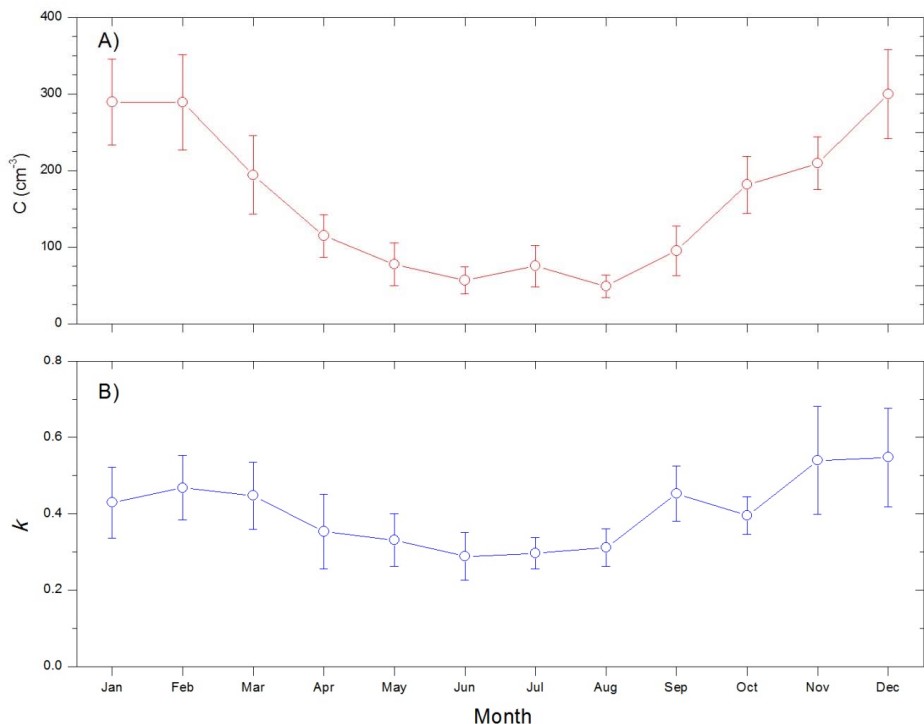

Figure 11. Seasonality of monthly mean values of (a) $C$ and (b) $k$ over the whole observation periods.





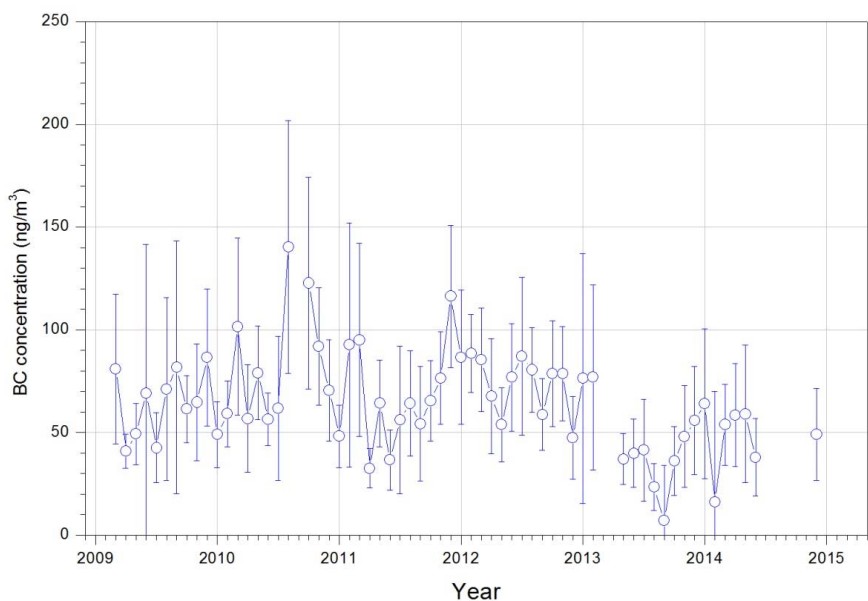

Figure 12. Monthly mean concentrations of black carbon over the period from March 2009 to February 2015.




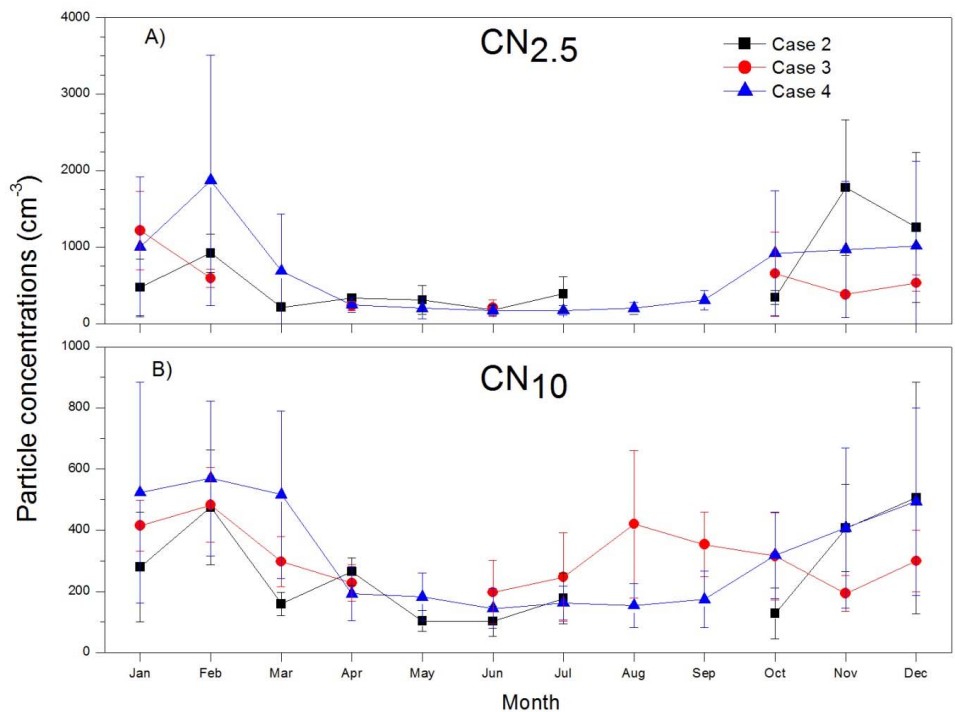

Figure 13. Seasonal variation of mean (a) $CN_{2.5}$ and (b) $CN_{10}$ concentrations with a standard deviation

depending on the air mass origin.





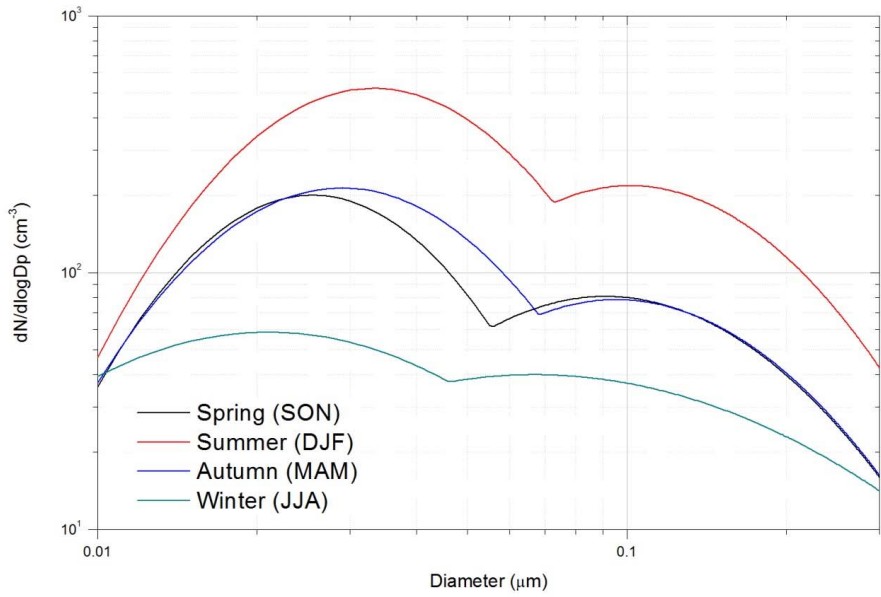

Figure 14. Seasonal lognormally fitted size distribution of aerosol particles originating from the South
5      Pacific Ocean, ranging from 0.01 to 0.3 μm (Case 4).



Table 1. Summary of meteorology and aerosol data according to the origin and transport pathway of aerosol particles. Case 1, Case 2, Case 3, and Case 4 refer to the origin and pathway of the air masses from South America, South Atlantic Ocean, Antarctica and South Pacific Ocean, respectively.

| | Overall | Case 1 | Case 2 | Case 3 | Case 4 |
|---|---|---|---|---|---|
| Wind speed (m s$^{-1}$) | 8.4 ± 1.8 | 2.6 ± 1.1 | 6.0 ± 1.5 | 6.7 ± 1.7 | 8.6 ± 1.8 |
| Wind direction (deg) | 237.2 ± 55.8 | 186.2 ± 20.7 | 155.9 ± 50.3 | 206.9 ± 52.3 | 242.7 ± 55.3 |
| BC concentrations (ng m$^{-3}$) | 65.1 ± 29.2 | 122.2 ± 10.6 | 36.7 ± 14.2 | 65.6 ± 30.0 | 66.5 ± 29.5 |
| CCN concentrations (cm$^{-3}$) | 129.7 ± 50.5 | 212.8 ± 50.2 | 146.0 ± 50.3 | 128.9 ± 34.9 | 128.7 ± 50.8 |
| CN$_{2.5}$ concentrations (cm$^{-3}$) | 737.3 ± 849.4 | 374.9 ± 64.4 | 605.3 ± 517.6 | 578.9 ± 377.3 | 751.2 ± 877.1 |
| CN$_{10}$ concentrations (cm$^{-3}$) | 347.8 ± 229.1 | 358.8 ± 61.2 | 268.8 ± 173.9 | 331.9 ± 133.0 | 352.2 ± 234.9 |
| Frequency | | 3 | 113 | 118 | 2407 |



Table 2. Seasonal size distribution lognormal fitting parameters for the Aitken and Accumulation mode of aerosol particles originating from a Case 4 scenario. N, σ, and $D_g$ refer to the number concentrations, a standard deviation, and the geometric mean diameter, respectively.

| | Aitken mode | | | Accumulation mode | | |
|---|---|---|---|---|---|---|
| | N (cm⁻³) | σ | $D_g$ (μm) | N (cm⁻³) | σ | $D_g$ (μm) |
| Spring (SON) | 112.010 | 1.655 | 0.026 | 53.873 | 1.939 | 0.094 |
| Summer (DJF) | 304.359 | 1.727 | 0.034 | 140.250 | 1.823 | 0.109 |
| Autumn (MAM) | 118.643 | 1.764 | 0.028 | 50.934 | 1.901 | 0.092 |
| Winter (JJA) | 49.164 | 2.296 | 0.023 | 44.780 | 2.827 | 0.086 |

