# Peer review of "Seasonal variations in physical characteristics of aerosol particles at the King Sejong Station, Antarctic Peninsula"

_Atmospheric Chemistry and Physics, 2016_

## Referee Comment (RC1) · Anonymous Referee #1 · 3 Nov 2016

**Paper: acp- 2016-795**

*Interactive comment on* " Seasonal variations in physical characteristics of aerosol particles at the King Sejong Station, Antarctic Peninsula", Kim et al.

**General remarks**

The authors report the measurements of aerosol number concentrations, cloud condensation nuclei (CCN), black carbon, and meteorological conditions over a six-year period.

The authors claim in the *Introduction* that "it is necessary to have long-term observations at different regions because aerosol particles vary temporally and spatially." It seems to me that the authors do not make full use of the data measured for a relatively long time (March 2009 to February 2015).

Given extensive data available, interest in the paper would be enhanced if the authors had obtained more significant results. For instance, in addition to a seasonal trend, the authors should check if an annual trend exists for aerosol number concentrations, CCN, and air temperature.

I will discuss in detail several points which need to be broadened, analysed and corrected.

**Specific coments**

**A) Page 5**

Line 12 and following:

The experimental part should be broadened highlighting the important points. The authors should clarify:

a) If the relative humidity (RH) of the sampled air was adjusted (e.g. at 40%) or not at the inlet of the SMPS.

b) The length and diameter of the main tube and the tubes connecting the stack with the sampling devices, showing if the flow is laminar or turbulent.

c) If the total counting efficiency of the system (main line and sampling lines) was computed.

d) If CPC and DMA were calibrated before and during the campaigns, which lasted about six years.

e) If particle, CN and CCN concentrations are shown in standard conditions.

Line 18 and following:

*"The aethalometer was used to measure the concentration of light absorption particles at two wavelengths (370 and 880 nm). In this study, we used the results obtained by measuring light absorption at 880 nm to determine the BC concentrations."*

Please insert the manufacturer of the aethalometer. The authors should clarify why they take into account aerosol absorption at 880 nm.

**B) Page 7**

Line 3 and following:

*" Fig.3 depicts monthly variations of the meteorological parameters measured from…*

The authors should discuss possible correlations between the considered parameters, and possible variations in these parameters (e.g. temperature trend) during the period considered (2009 – 2015).

Line 6 and following

*"…the observation site was relatively humid and warm condition compared to other Antarctic stations…"*

The statement should be changed to:

"..the observation site was relatively humid and warm compared to inland Antarctic stations..".

Line 16:
*"(DJF).....(JJA)"*
Should be changed to: ..."maximum in the summer (from December to February, DJF) and minimum in the winter (from June to August,  JJA)".

Line 20:
*"There are no significant anthropogenic sources of aerosol particles in Antarctica, therefore, our results were in good agreement... "*
The statement should be changed to: "Our results were in good agreement with the results...".

**C)  Page 8**
Line 2 and following:
*" The major compounds of aerosol particles found at a coastal Antarctic regions were non-sea salt sulphate and methane sulphonate (MSA) derived from oxidation of DMS produced by phytoplankton (Weller et al., 2011).*
Weller et al.'s conclusion (2011) is different. They write referring to Neumayer station*: "From thermodenuder experiments we deduced that the portion of volatile (at 125°C) and semi-volatile (at 250°C) particles which could be both associated with biogenic sulphur aerosol, was maximum during austral summer, while during winter non-volatile sea salt particles dominated."*.

Atmospheric marine aerosol consists prevalently of primary aerosol (organic material, sea-salt) produced on the ocean surface by bubble bursting and wave crest disruption, and biogenic secondary aerosol (non-sea-salt sulphate and methanesulphonic acid from oxidation of DMS emitted by phytoplankton, and ammonium from biological reduction processes of N-cycle compounds).
During the 2002-2003 summer  season, Fattori et al. (2005) reported that the coastal site ("Mario Zucchelli Station" in Terra Nova Bay) was affected by primary and secondary marine input: the sea spray contribution was dominant in the coarse fraction whereas the biogenic source prevailed in the fine fraction.

Line 8 and following
*"The CN concentrations typically increase in the summer due to high biological activity, while they decrease in the winter when biological activity is low...."*

The CN concentration is related not only to biological activity, but also to primary aerosol. Primary aerosol includes inorganic salts, inorganic and organic mixture, and biological particles.
The contribution of primary aerosol to the total aerosol concentration depends mainly on wind speed and the season, and it is higher in winter and lower in summer. In general, factors affecting total particle number concentrations are the air mass type, meteorological conditions, and whether or not nucleation-mode particles are present.

Line 14 and following
"O*ur results suggest that $CN_{2.5}$ concentrations may be more closely coupled with solar radiation intensity than with temperature"*.

The problem is more complex. An important parameter could interfere, i.e. the ocean temperature, which is different from air temperature. For instance, at the Antarctica research station Aboa Virkkula et al. (2009) observed that the annual maximum daily-averaged particle concentration was later, in February, than the maximum in solar radiation intensity. They concluded that the particle concentrations are more closely linked with the ocean temperature than with solar radiation. Peak sea temperature in polar regions is reached in late summer.
As the authors measured both solar radiation and CN concentration, it could be important to point out if there is a delay between the maximum CN and the maximum solar radiation.

In the summer 2014 the $CN_{2.5}$ and $CN_{10}$ concentrations are much lower compared to remaining considered years (Fig.4), but the solar radiation (Fig. 3) remains roughly stable during the summer in the years 2010-2015. These data should be explained.

Line 17 and following
" *The monthly mean CN concentrations increased from September to February…*"
Several papers report the annual variation of CN with the maximum in summer and the minimum in winter in coastal areas (Bigg et al., 1984; Gras and Adriaansen, 1985; Gras, 1993; Jaenicke et al, 1992) and in inland stations (Bigg et al., 1984; Samson et al., 1990). A few references should be cited.

**D) Page 9**
Line 10 and following:
*"The clear seasonality of CCN concentrations is probably caused by the seasonal trend of CN concentrations…"*
The statement should be changed to: The clear seasonality of CCN concentrations follows the trend of CN.

**E) Page 10**
Line 3 and following
*"The fraction at the SS value of 0.2% during the winter(JJA) was similar to those measured in Mace Head and Finokalia, which are regions representative of marine environment."*
*"Our observations suggests that the major components in the aerosol particles that are activated to CCN at an SS of 0.2% should be hygroscopic sea salts during winter, while compounds less hygroscopic than sea salt would be dominant during the summer."*.

Comparison with the Finokalia site appears inappropriate, as Bougiatioti et al. (2009) performed measurements at Finokalia from mid-June to mid-October (i.e. summer and autumn), not in winter. In addition, Bugiatioti et al. state that "Finokadia is located at a unique "crossroad" of aged aerosol types (marine boundary layer, Saharan desert, European sub-continent, biomass burning events) during the summer period." In the case of King Sejong Station, only a few cases of air masses originated from the continent of South America were shown. The authors' conclusion appears to be oversimplified and should be better explained. In addition to sea-salts, sea-spray aerosol includes organic material (prevalently water insoluble) which possesses a low hygroscopic growth factor, while simultaneously having a CCN activation efficiency higher than soluble non-sea-salt sulphate (Ovadnevaite et al., 2011).

Line 8 and following
*"Fig. 10 illustrates the seasonal variations in the mean activation ratio of CCN concentrations at an SS of 0.4%..."*
As in the previous paragraph, the authors discuss the trend of CCN concentration at SS= 0.2 %, I expect the authors considered the seasonal variations of the ratio between CCN and CN concentrations at supersaturation 0.2%, instead of 0.4%.

**F) Page 11**
Line 16 and following
*"The BC concentrations observed at our station were slightly higher than those at other stations in Antarctica…." "Additionally, no clear seasonal patterns were observed in our study throughout the entire observation period."*.
The BC concentrations measured by the authors are much higher than those at other Antarctic stations. Please compare the concentrations shown in the paper concerning South Pole, Halley,

Neumayer, and Ferraz station (0.65 ng m$^{-3}$, 1.0 ng m$^{-3}$, 2.6 ng m$^{-3}$ and 8.3 ng m$^{-3}$. respectively), with those measured at King Sejong Station. The very high concentration measured needs to be explained. In addition, it appears to me that a seasonal trend can be noted from Fig.12, i.e. prevalently lower values during winter.

**G) Page 12**
Line 6
*As mentioned earlier in Sec. 2.3*
Please change to: As mentioned earlier in Sec. 2.2

Line 8
*"Although they are unreliable due to the low observation frequency...."*
I suggest changing this statement to: " The very few cases of air masses originated from the continent of South America show the highest BC and CCN concentrations (Table 1).

**H) Page 14**

Line 3 and following
*"The activation fraction of aerosol particles at the King Sejong Station....was lower than at the Arctic sites indicating that less hygroscopic compounds in aerosol particles should be dominant"*.
No reference is shown for Arctic sites.
The conclusion appears superficial. I recall Ovadnevaite et al.'s paper (2011) which shows that sea-spray aerosol enriched in primary organic matter (prevalently hydrophobic) possesses more CCN activation efficiency than more soluble particles dominated by nss-sulphate.

Line 13
*"Although the BC and CCN concentrations were the highest when the air mass originated from the South American continent, the results are not significant because only a small amount of data was analyzed."*.
I suggest changing this to: " The very few cases of air masses originated from the continent of South America showed the highest BC and CCN concentrations."

**I) Page 21**
Figure 1 shows devices like OPC and Nephelometer, not used in the measurements.

**References**
Bigg et al., 1984. J. Atmos. Chem., 1, 203-214.
Bougiatioti et al., 2009. Atmos. Chem. Phys., 9, 7053-7066.
Fattori et al., 2005. J. Environ. Monit., 7, 1265-1274.
Gras and Adriaansen, 1985. J. Atmos. Chem., 3, 93-106.
Gras, J.L., 1993, Atmos. Environ., 9, 1417-1425.
Jaenicke et al., 1992, Tellus, 44B, 311-317.
Ovadnevaite et al., 2011. Geophys. Res. Letters, 38, L21806, doi:10.1029/2011GL048869.
Samson et al., 1990. Atmos. Res., 25, 385-396.
Virkkula et al.,  2009. Geophysica, 45, 163-181.
Weller et al., 2011. Atmos. Chem. Physics, 11, 13242-13257.

---

## Referee Comment (RC2) · Anonymous Referee #2 · 29 Nov 2016

This manuscript summarizes measurements of aerosol physical properties (size distributions and number concentrations) as well as CCN concentrations performed at a research station in Antarctica. These data are needed to establish quantitative points of reference for a region that may face dramatic changes as a result of climate change. Because of this, this manuscript fulfills an important role and is suitable for publication in ACP.

I do feel that the authors have missed some opportunities in their presentation of the data to dig a little deeper into their data. I would like for the authors to comment on some of the more obvious issues to me:

(1) Very little is mentioned about the particle size distributions. For example, if new particle formation is expected (and there are several references to suggest this in the text), then what do the SMPS data tell us about the nature of the particle formations events. To address this question the data from the size distribution shown in Fig. 8 could show "box-whisker" data that better account for less frequent new particle formation events.

(2) Another missed opportunity is the lack of a thorough analysis of the CCNC data. Other than reporting concentrations at specific supersaturations, the "spectrum" of CCN activity is not really discussed in this paper. Some fitting parameters, namely "C" and "k", are presented but one of the most widely used parameters, kappa from k-Koehler theory, is not even mentioned.

While I do not demand consideration of these issues as a condition for publication, I do urge the authors to consider how a somewhat minor investment in time might add great value to this publication.

The following are a list of edits and concerns, with the page and line numbers listed:

P2, line 5: lager -> larger

Line 10: This abstract should not assume that the reader understands what C and k refer to. If you want to include these in the abstract you need to explain what they are.

Line 17: improper use of semicolon. Replace with colon

P3, line 9: "unascertained" may be a word (I'm not sure) but many simpler words like "undetermined" or "unknown" are better suited.

Line 10: remove "their"

Line 21: e.g. not i.e. (this is not an exhaustive list of physical properties).

P6, line 1: "decided" should be replaced with "determined"

Line 2: delete the second "only"

Line 7: what model of DMA was used?

Line 12: "Besides" might be better replaced with "in addition" (this is clearly a style suggestion)

18: "were" should be "was"

P7, line 8: Please review this sentence and correct grammar

P9, Line 1: Please explain why, for this analysis, a supersaturation of 0.4% was chosen

Line 10: is this redundant with the previous sentence in line 5?

Line 19: any evidence of growth from Aitken to nucleation mode?

P10, Line 15: correct spelling of Arctic

P11, Line 5: how well did this model fit the data? There is no mention of this.

Line 17: period (not periods)

P13, Line 6: no comma after "for"

P14, Line 5: all mention of CCN concentrations need to state the SS

Line 10: again, I don't think the reader knows immediately what C and k refer to. If the authors want this section to summarize results I would suggest explaining this to the reader.

Figure notes: Figs 4,8, 10-13: what do the error bars represent? This needs to be in the caption.

Figure 9. Why not normalize to total N?

Fig 13: seems statistically the same to me.

Fig 14: needs to show something about the variability of the size distributions.

---

## Author Comment (AC1) · 28 Feb 2017

We thank the referee for valuable comments that we have used to improve our manuscript. We have considered the comments and have modified the manuscript accordingly. Our detailed responses to the referee's comments are below.

**General remarks**

The authors report the measurements of aerosol number concentrations, cloud condensation nuclei (CCN), black carbon, and meteorological conditions over a six-year period.

The authors claim in the Introduction that "it is necessary to have long-term observations at different regions because aerosol particles vary temporally and spatially." It seems to me that the authors do not make full use of the data measured for a relatively long time (March 2009 to February 2015).

Given extensive data available, interest in the paper would be enhanced if the authors had obtained more significant results. For instance, in addition to a seasonal trend, the authors should check if an annual trend exists for aerosol number concentrations, CCN, and air temperature.

I will discuss in detail several points which need to be broadened, analysed and corrected.

Authors' response: We did analysis for the annual trends of temperature, CN, and CCN concentrations as shown in Fig. S1-S3 of this response. As displayed in Fig. S1-S3, no clear annual trends of temperature, CN, and CCN concentrations are observed during the six-year period, mainly due to a relatively short observation period. For the analysis of long-term trends, authors reached a conclusion that longer term observations are needed, and not to include in the manuscript.

[Figure]

Figure S1. Box plot of annual variations of temperature during whole observation period. Lines in the middle of the boxes indicate sample medians (mean: circle), lower and upper lines of the boxes are the 25th and 75th percentiles, and whiskers indicate the 5th and 95th percentiles.

[Figure]

Figure S2. Box plot of annual variations of $CN_{2.5}$ and $CN_{10}$ concentrations. Lines in the middle of the boxes indicate sample medians (mean: circle), lower and upper lines of the boxes are the 25th and 75th percentiles, and whiskers indicate the 5th and 95th percentiles.

[Figure]

Figure S3. Box plot of annual trends of CCN concentrations. Lines in the middle of the boxes indicate sample medians (mean: circle), lower and upper lines of the boxes are the 25th and 75th percentiles, and whiskers indicate the 5th and 95th percentiles.

**Specific comments**

**A) Page 5**

Line 12 and following:

The experimental part should be broadened highlighting the important points. The authors should clarify:

a) If the relative humidity (RH) of the sampled air was adjusted (e.g. at 40%) or not at the inlet of the SMPS.

Authors' response: The aim of this study is investigate physical characteristics of aerosol particles in ambient condition. The RH controller was not used at the inlet of instruments.

To clarify that we did not use dehumidifier to readers, we added the following sentence in Page 5 Line 24:

*"To maintain the ambient condition, any drying system was not used during sampling."*

b) The length and diameter of the main tube and the tubes connecting the stack with the sampling devices, showing if the flow is laminar or turbulent.

Authors' response: Based on GAW aerosol measurements guidelines and recommendations, we installed cylindrical stainless common inlet. The diameter and length of the main common inlet were 0.1 m and 5.2 m, respectively. In order to calculate Reynolds number in the common inlet we used average values of air pressure and temperature. They were 98.8 kPa and -2.4 $^{o}$C, respectively. The Reynolds number in the main tube was 2388. It represents that the flow in the main common inlet is transition regime (2000<Re<4000). For sampling, short L-bend tube made of stainless steel was placed at center of the main common inlet. Sampling was done by connecting instruments and main common inlet using conductive tubing. Diameter and length of the conductive tubing connecting the stack with the sampling devices are 3/8 inches and 0.6 m.

In the revised manuscript, we added the following paragraph on Page 5 line 13 to clarify the sampling method:

*"Based on Global Atmosphere Watch (GAW) aerosol measurements guidelines and recommendations, we installed cylindrical stainless common inlet. The common inlet was placed on the roof of the observatory (Fig. 1). The diameter and length of the common inlet were 0.1 m and 5.2 m, respectively. In order to understand flow condition in the common inlet, Reynolds number was calculated. We used mean values of air temperature and pressure measured over the period from March 2009 to February 2015. The mean values of temperature and pressure were -2.4 $^{o}$C and 98.8 kPa, respectively. The total flow rate of sample air was maintained as 150 lpm. The Reynolds number in the common inlet was 2388. It represents that the flow in the common inlet is transition regime (2000 < Re <4000). For sampling, short L-bend tube made of stainless steel was placed at center of the common inlet. Instruments were connected with the common inlet using conductive tubing to minimize the particle losses. Diameter and length of the conductive tubing connecting the stack with the sampling devices are 3/8 inches and 0.6 m, respectively."*

c) If the total counting efficiency of the system (main line and sampling lines) was computed.

Authors' response: We estimated the total counting efficiency of the common inlet. First of all, we calculated inlet efficiency. We used average values of air pressure and temperature. They were 98.8 kPa and -2.4 $^{o}$C, respectively. The inlet efficiency of aerosol particle range from 2.5 nm to 5 µm was about 1 (Baron and Willeke, 2001; Hinds, 1999). Then, we calculated efficiency of transport loss. We considered diffusion and sedimentation for calculating the efficiency of transport loss. We ignored loss from thermophoresis and coagulation. The efficiency of the sedimentation loss for aerosol particles range from from 2.5 nm to 5 um was about 1. And the efficiency of diffusion loss of aerosol particles was 0.92 for 2.5 nm particles and was about 0.99 for larger than 10 nm particles. Thus, the

total counting efficiency of 2.5 nm particles was 0.92 in the common inlet system used in this study, whereas it for larger than 10 nm particles was about 1. All sampling line except for the common inlet was conductive tubing. The conductive tubing has been used to minimize the known particle loss. Authors think above mentioned total counting efficiency do not have to be included in the manuscript.

d) If CPC and DMA were calibrated before and during the campaigns, which lasted about six years.

Authors' response: We always have extra CPCs *in-situ* which were maintained and calibrated by the manufacturer. If CPC had problems during observation period, overwintering researchers replaced bad CPC with extra CPC. The CPC in bad condition was sent to the manufacturer for maintenance and calibration. Status of instruments was checked every day by overwintering crews. Overwintering researchers regularly measured flow rate of CPCs (UCPC 3776 and CPC 3772) and calibrated zero count test for particle counter at the observatory. If flow rate and status of instruments were weird, we eliminated data during that period to improve data quality. The DMA was cleaned and calibrated for flow rate of sample and sheath air.

e) If particle, CN and CCN concentrations are shown in standard conditions.

Authors' response: CN and CCN concentrations of aerosol particles in ambient condition were measured by instruments in good condition. We filtered data when there are instruments error and malfunction symptoms. The dataset used in this manuscript are believed to be reliable data measured with the well running CPCs and CCNC.

Line 18 and following:

"The aethalometer was used to measure the concentration of light absorption particles at two wavelengths (370 and 880 nm). In this study, we used the results obtained by measuring light absorption at 880 nm to determine the BC concentrations."

Please insert the manufacturer of the aethalometer. The authors should clarify why they take into account aerosol absorption at 880 nm.

Authors' response: In this study, we used AE-16 model manufactured from Magee Scientific. In the revised manuscript, we insert the aethalometer model accordingly. Although we got results at two wavelengths (370 nm and 880 nm) from the instrument, manufacturer recommended that results obtained by measuring near-infrared wavelength (880 nm) for analyzing BC concentrations. Data obtained from 370 nm wavelength were usually used to analyze aromatic organic species. Because data at 370 nm wavelength do not have enough sensitivity for analyzing BC concentrations, 370 nm

wavelength of the aethalometer has not been mentioned in the revised manuscript.

We modified text to following text in Page 6 Line 4:

  *"The aethalometer (Magee Scientific, AE16) was used to measure the concentration of light absorption particles at 880 nm wavelength."*

**B) Page 7**

Line 3 and following

"Fig.3 depicts monthly variations of the meteorological parameters measured from…

The authors should discuss possible correlations between the considered parameters, and possible variations in these parameters (e.g. temperature trend) during the period considered (2009 – 2015).

Authors' response: we added discussion of variations in temperature trend accordingly. As shown in Figure S1 in this response, the temperature variation does not have meaning due to relatively short period to verify the temperature trend. In addition, trend analysis of temperature and solar radiation are out of the scope of this manuscript. Instead of the trend analysis of temperature and solar radiation, we focused on correlation analysis between solar radiation and CN concentration, which is described in Figure 5 and section 3.2.1.

We added the following sentence at Page 7 Line 25:

*"No clear annual trends of temperature are observed during a six-year period due to a relatively short observation period. In this manuscript, we focused on correlation analysis between temperature (or solar radiation) and CN concentration."*

Line 6 and following

"…the observation site was relatively humid and warm condition compared to other Antarctic stations…"

The statement should be changed to: "..the observation site was relatively humid and warm compared to inland Antarctic stations..".

Authors' response: text was changed accordingly.

Line 16:

"(DJF)…..(JJA)"

Should be changed to: …"maximum in the summer (from December to February, DJF) and minimum in the winter (from June to August, JJA)".

Authors' response: text was changed accordingly.

Line 20:

"There are no significant anthropogenic sources of aerosol particles in Antarctica, therefore, our results were in good agreement… "

The statement should be changed to: "Our results were in good agreement with the results…".

Authors' response: text was changed accordingly.

**C) Page 8**

Line 2 and following:

"The major compounds of aerosol particles found at a coastal Antarctic regions were non-sea salt sulphate and methane sulphonate (MSA) derived from oxidation of DMS produced by phytoplankton (Weller et al., 2011).

Weller et al.'s conclusion (2011) is different. They write referring to Neumayer station: "From thermodenuder experiments we deduced that the portion of volatile (at 125°C) and semi-volatile (at 250°C) particles which could be both associated with biogenic sulphur aerosol, was maximum during austral summer, while during winter non-volatile sea salt particles dominated.".

Atmospheric marine aerosol consists prevalently of primary aerosol (organic material, sea-salt) produced on the ocean surface by bubble bursting and wave crest disruption, and biogenic secondary aerosol (non-sea-salt sulphate and methanesulphonic acid from oxidation of DMS emitted by phytoplankton, and ammonium from biological reduction processes of N-cycle compounds).

During the 2002-2003 summer season, Fattori et al. (2005) reported that the coastal site ("Mario Zucchelli Station" in Terra Nova Bay) was affected by primary and secondary marine input: the sea spray contribution was dominant in the coarse fraction whereas the biogenic source prevailed in the fine fraction.

Authors' response: We understand that the text in the 1$^{st}$ version manuscript was not clear enough, possible to mislead the intention what we wanted to explain. Following referee's suggestion, we changed the sentence as to make the meaning clearer. Weller's conclusion is very well acknowledged, in this manuscript, we mean that highly CN2.5 concentrations during the austral summer season (DJF) most likely to be related to nss-sulphate and MSA derived from oxidation of DMS produced by phytoplankton since secondary formation aerosols play an important role in CN2.5 concentration.

To make clear the intention and meaning, we changed sentence in Page 8 Line 15 as:

"The high CN$_{2.5}$ concentrations during the austral summer season (DJF) should be related to non-

*sea-salt sulphate and methanesulphonate (MSA) derived from oxidation of dimethyl sulphide (DMS)*
*produced by phytoplankton (Weller et al., 2011)."*

Line 8 and following

"The CN concentrations typically increase in the summer due to high biological activity, while they decrease in the winter when biological activity is low.…"

The CN concentration is related not only to biological activity, but also to primary aerosol. Primary aerosol includes inorganic salts, inorganic and organic mixture, and biological particles. The contribution of primary aerosol to the total aerosol concentration depends mainly on wind speed and the season, and it is higher in winter and lower in summer. In general, factors affecting total particle number concentrations are the air mass type, meteorological conditions, and whether or not nucleation-mode particles are present.

Authors' response: Our intention was that difference between CN2.5 and CN10 concentrations typically increased in the summer season, whereas those in the winter decreased. Our hypothesis is that the trends of difference should be related to secondary aerosol formation caused by biological activity. Because temperature and solar radiation play an important role in the biological activity, we focused on the correlation between CN2.5 concentration and temperature, and between CN2.5 concentration and solar radiation.

To clarify our intention, we changed text to following sentence in Page 8 Line 21:
*"The difference between $CN_{2.5}$ and $CN_{10}$ concentrations typically increased in the summer season (DJF) due to high biological activity, whereas those in the winter season (JJA) decreased when biological activity is low. Our hypothesis is that trends of the difference should be related to secondary aerosol formation caused by biological activity."*

Line 14 and following

"Our results suggest that CN2.5 concentrations may be more closely coupled with solar radiation intensity than with temperature".

The problem is more complex. An important parameter could interfere, i.e. the ocean temperature, which is different from air temperature. For instance, at the Antarctica research station Aboa Virkkula et al. (2009) observed that the annual maximum daily-averaged particle concentration was later, in February, than the maximum in solar radiation intensity. They concluded that the particle concentrations are more closely linked with the ocean temperature than with solar radiation. Peak sea temperature in polar regions is reached in late summer. As the authors measured both solar radiation

and CN concentration, it could be important to point out if there is a delay between the maximum CN and the maximum solar radiation.

Authors' response: Virkkula el al. (2009) compared daily average particle number concentration and solar radiation from December 2003 to April 2004. Compared with our data-set, it seems that their observation period is not long enough to clearly verify correlation between particle concentrations and solar radiation. Cayan (1980) reported that sea surface temperature and air temperature have roughly same variance. We assume that seasonal trend of ocean temperature should be similar to those of air temperature. We also compare monthly variations of CN2.5, temperature, and solar radiation, as can be seen in Figure S4 of this response. CN2.5 concentration sharply decreased from March, while temperature decrease occurs later, say in May, whereas solar radiation gradually decreased from February. Although temperature gradually decreased in the winter, in addition, CN2.5 concentrations were stable as well as solar radiation. Correlation between CN2.5 concentrations and temperature, and between CN2.5 concentrations and solar radiation would be different month by month. However, relationship among monthly mean values of CN2.5, temperature, and solar radiation was investigated and explained in this study (see Figure 5 and discussions of the manuscript).

[Figure]

Figure S4. A comparison of monthly variations of $CN_{2.5}$ concentrations, temperature, and solar radiation during whole observation period.

In the summer 2014 the CN2.5 and CN10 concentrations are much lower compared to remaining

considered years (Fig.4), but the solar radiation (Fig. 3) remains roughly stable during the summer in the years 2010-2015. These data should be explained.

Authors' response: The reasons for lower CN2.5 concentrations in the summer 2014 could not be explained by solar radiation and temperature because solar radiation and temperature did not show any distinctive variation compared with other years. Other metrological parameters such as wind, air pressure, and RH could not explain the lower CN2.5 concentrations neither. The possible reason is the type of air masses reached to the sampling site. As shown in Figure S5, Case IV where air mass was originated from the South Pacific Ocean was dominant in the summer. Remarkable results in the summer 2014 were that frequency of Case II (air mass was originated from the South Atlantic Ocean) was high and frequency of Case IV was lower than other years. Based on the air mass back trajectory analysis as explained in Sec 2.2 of the manuscript, frequency of four types of air mass were compared in summer season. Seasonal trends of CN2.5 concentrations were different according to air mass history as shown in Figure 13 in the manuscript. In case of Case II, peak CN2.5 concentrations were in November, while maximum CN2.5 concentrations of Case IV were in February. Therefore, it is that increasing frequency of air mass originated from the South Atlantic Ocean (case 2) would explain this lower CN2.5 concentration.

To explain lower CN2.5 concentrations during the 2013-2014 summer season, we added the fowling paragraph on Page 9 Line 7:

*"Unique results of $CN_{2.5}$ concentrations were observed as shown in Fig. 4. The $CN_{2.5}$ concentrations in the summer season of 2013-2014 were much lower than other years. Unfortunately, the reason for the lower $CN_{2.5}$ concentrations could not be explained by solar radiation intensity and temperature because the solar radiation and the temperature did not show any distinctive variation compared with other years. The possible reason is type of air masses reached to the sampling site. Although air mass originated from the South Pacific Ocean (Case IV:* descriptions of the cases I, II, III and IV are described in section 3.3) *was dominant in the summer, based on the air mass back trajectory analysis as explained in Sec 2.2, frequency of air mass originated from the South Atlantic Ocean (Case II) in the summer of 2013-2014 was higher than other years and frequency of air mass originated from Case IV was lower than other years. In case of Case II, peak $CN_{2.5}$ concentrations were in November, while maximum $CN_{2.5}$ concentrations of Case IV were in February. Therefore, it is*

*that increasing frequency of air mass originated from the South Atlantic Ocean would explain this lower CN$_{2.5}$ concentration."*

[Figure]

Figure S5. Frequency of air masses reached to sampling site depending on air mass history in the summer only

Line 17 and following

"The monthly mean CN concentrations increased from September to February…"

Several papers report the annual variation of CN with the maximum in summer and the minimum in winter in coastal areas (Bigg et al., 1984; Gras and Adriaansen, 1985; Gras, 1993; Jaenicke et al, 1992) and in inland stations (Bigg et al., 1984; Samson et al., 1990). A few references should be cited.

Authors' response: We add a few references.

**D) Page 9**

Line 10 and following:

"The clear seasonality of CCN concentrations is probably caused by the seasonal trend of CN concentrations…"

The statement should be changed to: The clear seasonality of CCN concentrations follows the trend of CN.

Authors' response: text was changed accordingly.

**E) Page 10**

Line 3 and following

"The fraction at the SS value of 0.2% during the winter(JJA) was similar to those measured in Mace Head and Finokalia, which are regions representative of marine environment."

"Our observations suggests that the major components in the aerosol particles that are activated to CCN at an SS of 0.2% should be hygroscopic sea salts during winter, while compounds less hygroscopic than sea salt would be dominant during the summer.".

Comparison with the Finokalia site appears inappropriate, as Bougiatioti et al. (2009) performed measurements at Finokalia from mid-June to mid-October (i.e. summer and autumn), not in winter. In addition, Bugiatioti et al. state that "Finokadia is located at a unique "crossroad" of aged aerosol types (marine boundary layer, Saharan desert, European sub-continent, biomass burning events) during the summer period." In the case of King Sejong Station, only a few cases of air masses originated from the continent of South America were shown. The authors' conclusion appears to be oversimplified and should be better explained. In addition to sea-salts, sea-spray aerosol includes organic material (prevalently water insoluble) which possesses a low hygroscopic growth factor, while simultaneously having a CCN activation efficiency higher than soluble non-sea-salt sulphate (Ovadnevaite et al., 2011).

Authors' response: Paramonov et al. (2015) compared results from CCNC measurements 14 sites around world. The sampling sites were just grouped according to location in their study. For instance, Finokalia, Mace Head and RHaMBLe campaign were representative sites for marine environment. However, Bougiatioti et al. (2009) measured physical and chemical characteristics of atmospheric aerosols according to air mass history during the short-term campaign at the Finokalia site. Although they showed different characteristics of aerosols depending to origin and pathway of air masses, it is not clear enough due to results from short-term measurements. To clarify, we have removed the comparison with Finokalia data in the revised manuscript.

In the revised manuscript, text was changed on Page 11 Line 14 as:

*"The fraction at the SS value of 0.2% during the winter (JJA) was similar to those measured in Mace Head, which is a representative site of a marine environment (Paramonov et al., 2015)."*

CCN activation efficiency depends on the constituents of aerosol particles (Dusek et al., 2006; Ovadnevaite et al., 2011). As only CCNC dataset are available in this study, there exists a limitation to infer the chemical compounds of aerosol particles. Thus, we discarded the citation of chemical

compounds (e.g. sea salts) of aerosol particles in the revised manuscript.

We changed sentence to following text in Page 11 Line 17:

*"Although CCN concentrations were low in the winter, our observations suggest that aerosol particles that are activated to CCN during the winter season should be more hygroscopic than those during the summer period."*

Line 8 and following

"Fig. 10 illustrates the seasonal variations in the mean activation ratio of CCN concentrations at an SS of 0.4%..."

As in the previous paragraph, the authors discuss the trend of CCN concentration at SS= 0.2 %, I expect the authors considered the seasonal variations of the ratio between CCN and CN concentrations at supersaturation 0.2%, instead of 0.4%.

Authors' response: The main purpose of this section is to see the seasonal variations of CCN activation ration at SS=0.4%.

To avoid confusion for readers, we divided the section on Page 11 Line 20:

*"3.2.3Activation ratio and Fitting parameter of CCN."*

**F) Page 11**

Line 16 and following

"The BC concentrations observed at our station were slightly higher than those at other stations in Antarctica…." "Additionally, no clear seasonal patterns were observed in our study throughout the entire observation period.".

The BC concentrations measured by the authors are much higher than those at other Antarctic stations. Please compare the concentrations shown in the paper concerning South Pole, Halley, Neumayer, and Ferraz station (0.65 ng m-3, 1.0 ng m-3, 2.6 ng m-3 and 8.3 ng m-3. respectively), with those measured at King Sejong Station. The very high concentration measured needs to be explained. In addition, it appears to me that a seasonal trend can be noted from Fig.12, i.e. prevalently lower values during winter.

Authors' response: The main scope of this manuscript is to understand seasonal trends of CN and CCN concentrations. Data from CN and CCN concentrations when BC concentrations were higher than 100 ng m-3 were discarded to improve data quality, while we used raw BC data for analysis of trend of BC concentrations. For this reason, mean BC concentrations were slightly high (64.68 ng m-

3). If we discard data when BC concentrations were higher than 100 ng m-3, the mean BC concentrations sharply decreased as 27.43 ng m-3 during whole observation period. However, mean BC concentrations in this study were higher than those measured at other Antarctic stations (e.g. South Pole, Halley, and Neumayer). The reason for the higher BC concentrations might be related to location of sampling site. There are nine permanent on-site stations on the Baton Peninsular of King George Island. In particular, six stations are located within a 10 km radius from the King Sejong Station. There should be extra bias of data from BC measurements due to effect of other stations.

We changed the Figure 12 and modified text of manuscript on Page 13 Line 3:

*"To eliminate effect of local pollution on observations, in this study, data where BC concentrations were higher than 100 ng m$^{-3}$, were discarded. The BC concentrations varied between 1.07 ng m$^{-3}$ and 75.97 ng m$^{-3}$, with a mean of 27.43 ± 4.98 ng m$^{-3}$."*

[Figure]

Figure 12. Monthly mean concentrations of black carbon over the period from March 2009 to February 2015. Here the error bars represents the standard deviation of the measurements from the mean value.

In the revised manuscript, we added fowling text to explain high BC concentration compared with results from other Antarctic station on Page 13 Line 9:

*"The reason of the higher BC concentrations might be related to location of sampling site. There are nine permanent on-site stations on the Baton Peninsula of King George Island. In particular, six*

*stations are located within a 10 km radius from the King Sejong Station. There should be extra bias of data from BC concentrations due to effect of other stations."*

As already mention of response, it is not main aim of this manuscript to understand BC trend. Data of BC concentrations were used to remove effect of local pollution on CN and CCN analysis. It would seem that there was seasonal trend of BC concentrations during short-term. For example, BC concentrations in the winter were lower than those in the summer during 2011. However, the BC concentrations in the winter were the highest in 2010 and no clear trend of BC concentrations was monitored during 2012. Because the variation of BC concentrations does not have meaning due to relatively short period to verify the BC trend, thus, we do not comment on seasonal trends of BC concentrations in this manuscript.

**G) Page 12**

Line 6

As mentioned earlier in Sec. 2.3

Please change to: As mentioned earlier in Sec. 2.2

Authors' response: Thank you for correction. Text was changed accordingly.

Line 8

"Although they are unreliable due to the low observation frequency…."

I suggest changing this statement to: "The very few cases of air masses originated from the continent of South America show the highest BC and CCN concentrations (Table 1).

Authors' response: text was changed accordingly.

**H) Page 14**

Line 3 and following

"The activation fraction of aerosol particles at the King Sejong Station….was lower than at the Arctic sites indicating that less hygroscopic compounds in aerosol particles should be dominant".

No reference is shown for Arctic sites. The conclusion appears superficial. I recall Ovadnevaite et al.'s paper (2011) which shows that seaspray aerosol enriched in primary organic matter (prevalently hydrophobic) possesses more CCN activation efficiency than more soluble particles dominated by nss-sulphate.

Authors' response: We add a reference in the manuscript. Lathem et al. (2013) showed CCN activation efficiency measured in Arctic area with aircraft during summertime. Based on physical

properties of aerosol particles, it is impossible to exactly understand chemical compounds of aerosol particles. Chemical characteristics of aerosol particles can be just deduced. We modified text to eliminate misunderstanding to readers.

In the revised manuscript, we modified text to following on Page 15 Line 26:

*"It suggests that aerosol particles in Antarctic Peninsula should be less hygroscopic than those in Arctic."*

Line 13

"Although the BC and CCN concentrations were the highest when the air mass originated from the South American continent, the results are not significant because only a small amount of data was analyzed.".

I suggest changing this to: "The very few cases of air masses originated from the continent of South America showed the highest BC and CCN concentrations."

Authors' response: We changed a sentence as referee's suggestion.

*"The very few cases of air masses originated from the South American continent showed the highest BC and CCN concentrations."*

**I) Page 21**

Figure 1 shows devices like OPC and Nephelometer, not used in the measurements.

Authors' response: In this manuscript, we didn't show date form OPC and Nephelometer. We modified Figure 1 to reduce confusion.

[Figure]

CPC: condensation particle counter
SMPS: scanning mobility particle sizer
CCNC: cloud condensation nuclei counter

Figure 1. A schematic diagram for the observation methods used in this study.

References

Baron and Willeke (B&W) Aerosol Measurement, 2nd Edition, J Wiley and Sons, 2001.

Bougiatioti, A., Fountoukis, C., Kalivitis, N., Pandis, S. N., Nenes, A., and Mihalopoulos, N.: Cloud condensation nuclei measurements in the marine boundary layer of the eastern Mediterranean: CCN closure and droplet growth kinetics, Atmos. Chem. Phys., 9, 7053-7066, 2009.

Cayan, D. R.: Large-scale relationships between sea surface temperature and surface air temperature, Mon. Weather Rev., 108, 1293-1301, 1980.

Hinds, Aerosol Technology, 2nd edition, J. Wiley and Sons, 1999.

Paramonov, M., Kerminen, V. M., Gysel, M., Aalto, P. P., Andreae, M. O., Asmi, E., Baltensperger, U., Bougiatioti, A., Brus, D., Frank, G. P., Good, N., Gunthe, S. S., Hao, L., Irwin, M., Jaatinen, A., Jurányi, Z., King, S. M., Kortelainen, A., Kristensson, A., Lihavainen, H., Kulmala, M., Lohmann, U., Martin, S. T., McFiggans, G., Mihalopoulos, N., Nenes, A., O'Dowd, C. D., Ovadnevaite, J., Petäjä, T., Pöschl, U., Roberts, G. C., Rose, D., Svenningsson, B., Swietlicki, E., Weingartner, E., Whitehead, J., Wiedensohler, A., Wittbom, C., and Sierau, B.: A synthesis of cloud condensation nuclei counter (CCNC) measurements within the EUCAARI network, Atmos. Chem. Phys., 15, 12211-12229, 10.5194/acp-15-12211-2015, 2015.

---

## Author Comment (AC2) · 28 Feb 2017

We thank the referee for valuable comments that we have used to improve our manuscript. We have considered the comments and have modified the manuscript accordingly. Our detailed responses to the referee's comments are below.

This manuscript summarizes measurements of aerosol physical properties (size distributions and number concentrations) as well as CCN concentrations performed at a research station in Antarctica. These data are needed to establish quantitative points of reference for a region that may face dramatic changes as a result of climate change. Because of this, this manuscript fulfills an important role and is suitable for publication in ACP.

I do feel that the authors have missed some opportunities in their presentation of the data to dig a little deeper into their data. I would like for the authors to comment on some of the more obvious issues to me:

(1) Very little is mentioned about the particle size distributions. For example, if new particle formation is expected (and there are several references to suggest this in the text), then what do the SMPS data tell us about the nature of the particle formations events. To address this question the data from the size distribution shown in Fig. 8 could show "box-whisker" data that better account for less frequent new particle formation events.

Authors' response: The main purpose is to understand general physical characteristics of aerosol particles at the King Sejong Station in Antarctic Peninsula. Because the study on new particle formation events is out of scope, in the manuscript, brief description about the particle size distributions was done and the particle size distribution data were used to support explanation for monthly trends of CCN concentrations as shown in Fig. 8. In accordance with referee's suggestion, we also showed "box-whisker" data of particle size distribution as shown in Fig. S1. Although there is much outlier in summer season, we don't know that it

indicates frequency of new particle formation events. We are preparing other manuscript related to new particle formation events and are analyzing deeply and carefully about results of the particle size distribution.

[Figure]

Figure S1. Box-Whisker plot of particle size distribution during (a) spring, (b) summer, (c) autumn, and (d) winter season. Red bars in the boxes indicate median values (mean values: green circle), whiskers indicate the 5[th] and 95[th] percentiles, and red cross out of whiskers represents outliers.

(2) Another missed opportunity is the lack of a thorough analysis of the CCNC data. Other than reporting concentrations at specific supersaturations, the "spectrum" of CCN activity is not really discussed in this paper. Some fitting parameters, namely "C" and "k", are presented but one of the most widely used parameters, kappa from k-Koehler theory, is not even mentioned. While I do not demand consideration of these issues as a condition for publication, I do urge the authors to consider how a somewhat minor investment in time might add great value to this publication.

Authors' response: If chemical constituents of aerosol particles are size-dependent, it is complicated to estimate CCN concentrations by using particle size distribution. Because size-dependent chemical information of aerosol particles is not available in this study, an

empirical parameterization using in-situ CCN measurements was used in this study. The CCN concentrations as a function of SS were fitted with an equation of the form $N_{ccn}=C(SS)^k$, where $N_{ccn}$ is CCN concentration at a certain SS value, and $C$ and $k$ are the fitting parameters. The fitting was done separately for each SS cycle of CCNC data. Besides that we have focused and found the fitting values c and k values, we tested single hygroscopicity parameter, kappa in κ-Köhler theory, as suggested by referee. We roughly estimated kappa by using monthly mean CCN concentration at SS of 0.4% and monthly mean total particle concentrations obtained from SMPS data. The kappa is approximately 1.18 in December. The possible reason for the high kappa value would be explained by limitations of measurements available. For example, we decided the critical diameter ($d_{crit}$) by comparing CCN concentration with the integrated SMPS data. In addition, CCN concentrations were measured without size-selection. We decided not to mention the kappa value in the revised manuscript.

The following are a list of edits and concerns, with the page and line numbers listed:

P2, line 5: lager -> larger

Authors' response: text was corrected accordingly.

Line 10: This abstract should not assume that the reader understands what C and k refer to. If you want to include these in the abstract you need to explain what they are.

Authors' response: We agree with referee's opinion. To help reader's understanding, text was added to explain what the $C$ and $k$ are.

We added following sentence on Page 2 Line 9:

*"Based on measured CCN data at each supersaturation ratio (SS),* empirical parameterization were also fitted using formula expressed by power-law function ($N_{ccn}=C\times(SS)^k$), where $N_{ccn}$ is the CCN concentrations at a given SS, and $C$ and $k$ are the fitting parameters."

Line 17: improper use of semicolon. Replace with colon

Authors' response: text was corrected accordingly.

P3, line 9: "unascertained" may be a word (I'm not sure) but many simpler words like "undetermined" or "unknown" are better suited.

Authors' response: text was corrected accordingly.

Line 10: remove "their"

Authors' response: text was corrected accordingly.

Line 21: e.g. not i.e. (this is not an exhaustive list of physical properties).

Authors' response: text was corrected accordingly.

P6, line 1: "decided" should be replaced with "determined"

Authors' response: text was corrected accordingly.

Line 2: delete the second "only"

Authors' response: text was corrected accordingly.

Line 7: what model of DMA was used?

Authors' response: We used a cylindrical DMA similar to the TSI 3081 model. The length, inner diameter, and outer diameter of the DMA were 44.42 cm, 0.953 cm and 1.905 cm, respectively.

We added following sentence in Page 6 Line 17:

*"The length, inner diameter, and outer diameter of the DMA were 44.42 cm, 0.953 cm, and 1.905 cm, respectively."*

Line 12: "Besides" might be better replaced with "in addition" (this is clearly a style suggestion)

Authors' response: text was corrected accordingly.

18: "were" should be "was"

Authors' response: text was corrected accordingly.

P7, line 8: Please review this sentence and correct grammar

Authors' response: text was corrected accordingly.

*"The solar radiation varied from 2.3 W m$^{-2}$ to 375.4 W m$^{-2}$, with a mean value of 81.2 ± 38.9 W m$^{-2}$."*

P9, Line 1: Please explain why, for this analysis, a supersaturation of 0.4% was chosen

Authors' response: Anttila et al. (2012) measured cloud droplet number concentration (CDNC) and CCN concentration at five different SS values (0.2, 0.4, 0.6, 0.8, and 1.0%) during the third Palls Cloud Experiment (PaCE-3). The campaign was performed from 11 September to 11 October, 2009. According to their results, they found high correlation between CDNC and CCN concentrations at a supersaturation of 0.4%. The CCN concentrations at less 0.4% supersaturation were lower than CDNC, while the CCN concentrations at higher than 0.4% supersaturation were higher than CDNC. Based on this result, variation of CCN concentrations at the supersaturation of 0.4% was analyzed in this study.

We added following sentence on Page 10 Line 7:

*"Anttila et al. (2012) measured cloud droplet number concentration (CDNC) and CCN concentrations at five SS values (0.2, 0.4, 0.6, 0.8, and 1.0%) during the third Palls Cloud Experiment (PaCE-3). They showed correlation between CDNC and CCN concentrations at each supersaturation. The relationship between CDNC and CCN concentrations at the SS value of 0.4% was approximately linear, while CCN concentrations were lower than CDNC when the SS value was lower than 0.4% and CCN concentrations at upper 0.4% higher than CDNC. Based on this result, in this study, the supersaturation of 0.4% was chosen to investigate seasonal variations of CCN."*

Line 10: is this redundant with the previous sentence in line 5?

Authors' response: We agree with referee's opinion. Because meaning of two sentences was similar, in same paragraph, we removed the sentence in line 5.

In the manuscript, we remove following sentence in Page 9 Line 5:

*"It was similar to the seasonal cycle of the CN concentrations."*

Line 19: any evidence of growth from Aitken to nucleation mode?

Authors' response: If the referee raises an issue if there is any evidence of growth from Aitken mode particle to CCN size, we cannot provide any direct evidence of this. Nevertheless, we compared monthly mean particle concentration measured from SMPS with CCN concentrations. The calculations showed that the number of accumulation mode particles cannot explain the measured CCN concentrations. It means that the rest of CCN are from the growth of smaller than accumulation mode particles, say from the Aitken mode, or residuals of cloud process. We could not provide direct evidence to distinguish these processes, we modified the sentence as:

Page 11, line1 has been changed to
*"Accumulation mode particles can easily act as CCN (Dusek et al., 2006), hence CCN concentrations increase during the summer and decrease during the winter."*

P10, Line 15: correct spelling of Arctic
Authors' response: Thanks. We checked spelling and changed.

P11, Line 5: how well did this model fit the data? There is no mention of this.
Authors' response: We estimated $C$ and $k$ values by using daily mean CCN concentrations at each SS value. The average correlation coefficient, r, was 0.978. We think it was a good fit.

In the revised manuscript, we added following sentence in Page 12 Line 18:
*"The average correlation coefficient, r, was 0.978."*

Line 17: period (not periods)
Authors' response: text was corrected accordingly.

P13, Line 6: no comma after "for"
Authors' response: text was corrected accordingly.

P14, Line 5: all mention of CCN concentrations need to state the SS
Authors' response: In summary section, we missed SS values for explaining CCN concentrations. It can give readers confusion. Thus, we modified sentence in the manuscript.

To clarify we modified sentence to following text on Page 15 Line 22:

*"In addition, we presented the clear seasonal trends of CCN concentrations at the supersaturation of 0.4%."*

Line 10: again, I don't think the reader knows immediately what C and k refer to. If the authors want this section to summarize results I would suggest explaining this to the reader.

Authors' response: We agree with referee's opinion.

To clarify meaning of *C* and *k*, we added following sentence on Page 16 Line 3:

*"The C and k are constants were estimated using approximate formula expressed by a power-law function ($N_{CCN}=C \times (SS)^k$) (Twomey 1959)."*

Figure notes: Figs 4, 8, 10-13: what do the error bars represent? This needs to be in the caption.

Authors' response: The error bars represent a standard deviation.

We add caption in Figures 4, 8, 10-13.

*"Here the error bars represent the standard deviation of the measurements from the mean value."*

Figure 9. Why not normalize to total N?

Authors' response: To investigate seasonal variations of fractions of CCN concentration at each SS value in aerosol particles activated CCN at a SS of 1.0%, we normalized it to CCN concentrations at a SS of 1.0% without total N.

Fig 13: seems statistically the same to me.

Authors' response: In this Figure, we found that seasonality of $CN_{2.5}$ concentrations were different in accordance with the air mass history. For instance, the $CN_{2.5}$ concentrations originating from the South Atlantic (Case II) were the highest in November, whereas the $CN_{2.5}$ concentrations originating from the South Pacific (Case IV) were the highest in February as can be seen in Figure 13. This is probably due to difference in chemical compounds that contributed to aerosol formation processes and/or in variations of biogenic activity according to the origin and transport pathway of air masses. This analysis has been

explained in the section 3.3. Unfortunately, we don't have chemical data of aerosol particles depending on air mass. To verify our hypothesis, further studies on chemical compositions of aerosol particles need to be carried out in the future.

Fig 14: needs to show something about the variability of the size distributions.
Authors' response: we showed the variation of modal diameter and number concentrations of the size distribution as can be seen in Table 2. In the revised manuscript, the following sentence on Page 14 Line 20 was mentioned.

*"The modal diameters with standard deviation and number concentrations are summarized in Table 2. It is obvious that the modal diameters during the summer are larger than those during the winter for both Aitken and accumulation modes: 0.023 µm in the winter and 0.034 µm in the summer for the Aitken mode and 0.086 µm in the winter and 0.109 µm in the summer for the accumulation mode. The number concentrations for the summer are also higher than the value for the winter for the Aitken and accumulation modes, 49.16 ± 3.88 cm$^{-3}$ during the winter and 304.36 ± 20.10 cm$^{-3}$ during the summer for the Aitken mode and 44.78 ± 14.24 cm$^{-3}$ in the winter and 140.25 ± 10.64 cm$^{-3}$ in the summer for the accumulation mode."*

Reference
Anttila, T., Brus, D., Jaatinen, A., Hyvärinen, A. P., Kivekäs, N., Romakkaniemi, S., Komppula, M., and Lihavainen, H.: Relationships between particles, cloud condensation nuclei and cloud droplet activation during the third Pallas Cloud Experiment, Atmos. Chem. Phys., 12, 11435-11450, 10.5194/acp-12-11435-2012, 2012.

---

## Referee Report (RR1)

The authors have addressed most of my concerns. I still have one minor and one major concern.

1) Minor concern: Figure S1, as presented to me in the authors' response, is unreadable. I have not been able to check the supplemental document but if it looks the same the authors need to make it clearer. Just to be perfectly clear, this is what I think a box-whisker plot should look like (I took this random plot from the web):

[Figure]

2) Major concern: The authors' response to my comment on the CCN measurements does not give me great comfort that the authors have made a careful analysis of their data. While I am happy that the authors took the time to calculate kappa, I would argue two issues:

(a) It is, in fact, quite common to estimate kappa based on bulk measurements of CCN concentrations and SMPS-derived size distributions. As the authors point out, an internal mixture must be assumed. While it's true that this may introduce uncertainties, it is still an important and in fact is an often utilized, method for estimating kappa. Just a quick literature search uncovered three studies where this was done [1-3], and I am sure there are dozens more.

(b) If the authors feel that they have performed a correct calculation of kappa then they should clearly state their methods and result in their manuscript. It is OK if the value of kappa is unreasonable (and 1.18 is an unreasonable value!!). But it is not OK (in my opinion) to perform a good calculation that provides an unsatisfactory answer and then ignore the result.

**References**

1.    Furutani, H., et al., *Assessment of the relative importance of atmospheric aging on CCN activity derived from field observations.* Atmospheric Environment, 2008. **42**(13): p. 3130-3142.
2.    Chang, R.Y.W., et al., *Comparison between measured and predicted CCN concentrations at Egbert, Ontario: Focus on the organic aerosol fraction at a semi-rural site.* Atmospheric Environment, 2007. **41**(37): p. 8172-8182.
3.    Jaatinen, A., et al., *The third Pallas Cloud Experiment: Consistency between the aerosol hygroscopic growth and CCN activity.* Boreal Environment Research, 2014. **19**: p. 368-382.

---

## Referee Report (RR2)

Review of the manuscript entitled "Seasonal variations in physical characteristics of aerosol particles at the King Sejong Station, Antarctic Peninsula" by J. Kim, Y. J. Yoon, Y. Gim, H. J. Kang, J. H. Choi, and B. Y. Lee, with reference no.: **acp-2016-795**.

This manuscript presents and analyses measurements of aerosol properties (number concentrations, size distributions), cloud condensation nuclei (CCN) and black carbon concentrations for a six-year period (March 2009 to February 2015) at the King Sejong research station in Antarctica.

The Cryosphere and especially Arctic and Antarctica are key components of the Earth's system, and are inherently sensitive to a changing climate serving as the most stunning indicators of climate change. On the other hand, among climate change drivers, aerosols still contribute the largest uncertainty to the total climate forcing estimate especially through the aerosol – cloud interactions. This is due to the great variety of aerosol types, both natural and anthropogenic, their short atmospheric lifetimes and to the subsequent high spatiotemporal variability of their physical and optical properties. The Antarctic continent being the most remote area on the planet from other continents and thus from anthropogenic activities and emissions, it is an ideal place for studying natural aerosol processes in order to understand them and to correctly distinguish between natural and anthropogenic factors influencing the climate. Apart from some long-range transported pollution aerosols, primary aerosol sources like mineral dust, vegetation, soot or secondary aerosols from gas to particle conversion are virtually absent on this almost completely ice-covered continent. Hence, marine air masses advected from the Southern Ocean surrounding the continent, remains the dominant source to the Antarctic aerosol load. Therefore, any dataset of original and accurate measurements that helps to elucidate physical processes taking place in a such climatically sensitive region is important.

In this framework the submitted manuscript is interesting and relevant to the topics of ACP. Moreover it is well written and organised and thus it could be published in the ACP Journal after taking into account the following comments.

The manuscript presents interesting results on Antarctic aerosols based on continuous relatively long term (six years) observations at King Sejong station. The dataset is unique and the analysis of measurements is quite adequate. Core of the manuscript is data analysis on a seasonal basis and at a next level according to the origin of air masses though timeseries of monthly mean data of some variables are presented (e.g. Fig. 3, Fig. 4, Fig. 7a and Fig. 12). Authors analyse the intra-annual variation of examined parameters and they discuss the main features of their seasonal behavior making an effort to provide possible explanations to interpret their findings. In some cases, they compare their results with that of other research works conducted in Antarctica. However, the whole analysis doesn't go deeper to gain an important insight into the factors determining the aerosol properties seasonality and the factors affecting the CCN activation. For instance, is the new particle formation the only or the main factor that induces seasonal variation in particle concentration (total and CN2.5)?

My main concern is about the gain of the new knowledge that this paper brings. Authors cite in the text, especially in the introduction, several works on Antarctic aerosols and their properties. So, what is the contribution of this manuscript to this knowledge? In the introduction authors state "*Although various studies have been performed, the measurements taken at the Antarctic Peninsula and the long-term observations of aerosol particles are still insufficient*" but they do not discuss any inter-annual variability or trend (except for the exceptional year of 2014). They focus on the seasonal variation. In order to support their work, authors should clearly state what are the new approach, analysis and/or findings compared to literature and this should be clearly presented in the concluding section as well. Otherwise, they can discuss their results compared to other works, examining whether they are in agreement strengthening thus the existing knowledge since current results are issued from multi-year observations. Actually they do it sometimes. For instance, authors state that the revealed seasonal pattern of $CN_{2.5}$ and $CN_{10}$ is in agreement with the results of previous studies (page 8, lines 8-10). I am wondering if the consideration of the seasonal variation of $CN_{2.5}$ and $CN_{10}$ separately, is additional and further information compared to previous works. On the other hand, throughout the discussion regarding timeseries, seasonal behavior of CCN concentrations, particles size distribution and CCN activation ratios, there are no references to other relative

studies in Antarctica. If this analysis and its findings give new or additional information should be stated by authors adding thus value to their work. The same is valid for the analysis regarding the effects of air masses origin on the aerosols physical properties.

I should however state that the lack of new knowledge doesn't reduce the value of a dataset of original measurements of aerosol properties with a relatively long temporal coverage, in a remote, not easily accessible and very interesting from climate change point of view, area of the planet.

**Some minor remarks**
- Authors give enough information about instrumentation but they do not discuss any quality control assessment that they apply to their raw records.
- Authors trying to interpret the exceptional CN concentration levels during the period 2013-2014, found that air masses origin was differentiating this period compared to previous years. Air masses from south Atlantic were more frequent than other years. A comparison of CN concentration levels with analogous measurements (if there are published) at stations which are affected mostly by south Atlantic air masses could support this argument.
- Analyzing the CCN concentration, it was found that its seasonal variation follows the seasonal cycle of particles concentration which is logical. I have however point out that the CCN concentrations during the period 2013-2014 seem to be unaffected by the low particles concentration in that period as they remain similar to other years.
- Page 3, lines 8-9. In the sentence *".... the direct and indirect climate effects are still unknown (IPCC, 2013).",* I think the word "unknown" is not appropriate. Actually, according to IPCC report, aerosol effects contribute the largest uncertainty in the total radiative forcing. Thus you can replace the word '*unknown'* by 'highly uncertain'.
- Page 3, line 21. In the sentence *"For these reasons, the observation of **the** physical properties in Antarctica, ...",* replace the word '*the'* by 'their'

- Page 7, line 14. In the sentence "*Fig. 3 depicts monthly variations of the meteorological parameters measured from **and** automatic weather system (AWS) ...*" replace the word '*and*' by 'an'

- Page 7, line 18. In the sentence "the observation site was relatively humid and warm **condition** compared to inland Antarctic stations", remove the word 'condition'

- Page 15, line 6. In the sentence "*Our results are **similar** those of previous laboratory and field experiments (Sellegri et al., 2006; Yoon et al., 2007).*", add the word '**to'** after the word '*similar*'.

---

## Author Response (ED1)

We thank the referee for valuable comments that we have used to improve our manuscript. We have considered the comments and have modified the manuscript accordingly. Our detailed responses to the referee's comments are below.

**General remarks**

The authors report the measurements of aerosol number concentrations, cloud condensation nuclei (CCN), black carbon, and meteorological conditions over a six-year period.

The authors claim in the Introduction that "it is necessary to have long-term observations at different regions because aerosol particles vary temporally and spatially." It seems to me that the authors do not make full use of the data measured for a relatively long time (March 2009 to February 2015).

Given extensive data available, interest in the paper would be enhanced if the authors had obtained more significant results. For instance, in addition to a seasonal trend, the authors should check if an annual trend exists for aerosol number concentrations, CCN, and air temperature.

I will discuss in detail several points which need to be broadened, analysed and corrected.

Authors' response: We did analysis for the annual trends of temperature, CN, and CCN concentrations as shown in Fig. S1-S3 of this response. As displayed in Fig. S1-S3, no clear annual trends of temperature, CN, and CCN concentrations are observed during the six-year period, mainly due to a relatively short observation period. For the analysis of long-term trends, authors reached a conclusion that longer term observations are needed, and not to include in the manuscript.

[Figure]

Figure S1. Box plot of annual variations of temperature during whole observation period. Lines in the middle of the boxes indicate sample medians (mean: circle), lower and upper lines of the boxes are the 25th and 75th percentiles, and whiskers indicate the 5th and 95th percentiles.

[Figure]

Figure S2. Box plot of annual variations of $CN_{2.5}$ and $CN_{10}$ concentrations. Lines in the middle of the boxes indicate sample medians (mean: circle), lower and upper lines of the boxes are the 25th and 75th percentiles, and whiskers indicate the 5th and 95th percentiles.

[Figure]

Figure S3. Box plot of annual trends of CCN concentrations. Lines in the middle of the boxes indicate sample medians (mean: circle), lower and upper lines of the boxes are the 25th and 75th percentiles, and whiskers indicate the 5th and 95th percentiles.

**Specific comments**

**A) Page 5**

Line 12 and following:

The experimental part should be broadened highlighting the important points. The authors should clarify:

a) If the relative humidity (RH) of the sampled air was adjusted (e.g. at 40%) or not at the inlet of the SMPS.

Authors' response: The aim of this study is investigate physical characteristics of aerosol particles in ambient condition. The RH controller was not used at the inlet of instruments.

To clarify that we did not use dehumidifier to readers, we added the following sentence in Page 5 Line 24:

*"To maintain the ambient condition, any drying system was not used during sampling."*

b) The length and diameter of the main tube and the tubes connecting the stack with the sampling devices, showing if the flow is laminar or turbulent.

Authors' response: Based on GAW aerosol measurements guidelines and recommendations, we installed cylindrical stainless common inlet. The diameter and length of the main common inlet were 0.1 m and 5.2 m, respectively. In order to calculate Reynolds number in the common inlet we used average values of air pressure and temperature. They were 98.8 kPa and -2.4 $^o$C, respectively. The Reynolds number in the main tube was 2388. It represents that the flow in the main common inlet is transition regime (2000<Re<4000). For sampling, short L-bend tube made of stainless steel was placed at center of the main common inlet. Sampling was done by connecting instruments and main common inlet using conductive tubing. Diameter and length of the conductive tubing connecting the stack with the sampling devices are 3/8 inches and 0.6 m.

In the revised manuscript, we added the following paragraph on Page 5 line 13 to clarify the sampling method:

*"Based on Global Atmosphere Watch (GAW) aerosol measurements guidelines and recommendations, we installed cylindrical stainless common inlet. The common inlet was placed on the roof of the observatory (Fig. 1). The diameter and length of the common inlet were 0.1 m and 5.2 m, respectively. In order to understand flow condition in the common inlet, Reynolds number was calculated. We used mean values of air temperature and pressure measured over the period from March 2009 to February 2015. The mean values of temperature and pressure were -2.4 $^o$C and 98.8 kPa, respectively. The total flow rate of sample air was maintained as 150 lpm. The Reynolds number in the common inlet was 2388. It represents that the flow in the common inlet is transition regime (2000 < Re <4000). For sampling, short L-bend tube made of stainless steel was placed at center of the common inlet. Instruments were connected with the common inlet using conductive tubing to minimize the particle losses. Diameter and length of the conductive tubing connecting the stack with the sampling devices are 3/8 inches and 0.6 m, respectively."*

c) If the total counting efficiency of the system (main line and sampling lines) was computed.

Authors' response: We estimated the total counting efficiency of the common inlet. First of all, we calculated inlet efficiency. We used average values of air pressure and temperature. They were 98.8 kPa and -2.4 $^o$C, respectively. The inlet efficiency of aerosol particle range from 2.5 nm to 5 μm was about 1 (Baron and Willeke, 2001; Hinds, 1999). Then, we calculated efficiency of transport loss. We considered diffusion and sedimentation for calculating the efficiency of transport loss. We ignored loss from thermophoresis and coagulation. The efficiency of the sedimentation loss for aerosol particles range from from 2.5 nm to 5 um was about 1. And the efficiency of diffusion loss of aerosol particles was 0.92 for 2.5 nm particles and was about 0.99 for larger than 10 nm particles. Thus, the

total counting efficiency of 2.5 nm particles was 0.92 in the common inlet system used in this study, whereas  for larger than 10 nm particles was about 1. All sampling line except for the common inlet was conductive tubing. The conductive tubing has been used to minimize the known particle loss. Authors think above mentioned total counting efficiency do not have to be included in the manuscript.

d) If CPC and DMA were calibrated before and during the campaigns, which lasted about six years.

Authors' response: We always have extra CPCs *in-situ* which were maintained and calibrated by the manufacturer. If CPC had problems during observation period, overwintering researchers replaced bad CPC with extra CPC. The CPC in bad condition was sent to the manufacturer for maintenance and calibration. Status of instruments was checked every day by overwintering crews. Overwintering researchers regularly measured flow rate of CPCs (UCPC 3776 and CPC 3772) and calibrated zero count test for particle counter at the observatory. If flow rate and status of instruments were weird, we eliminated data during that period to improve data quality. The DMA was cleaned and calibrated for flow rate of sample and sheath air.

e) If particle, CN and CCN concentrations are shown in standard conditions.

Authors' response: CN and CCN concentrations of aerosol particles in ambient condition were measured by instruments in good condition. We filtered data when there are instruments error and malfunction symptoms. The dataset used in this manuscript are believed to be reliable data measured with the well running CPCs and CCNC.

Line 18 and following:

"The aethalometer was used to measure the concentration of light absorption particles at two wavelengths (370 and 880 nm). In this study, we used the results obtained by measuring light absorption at 880 nm to determine the BC concentrations."

Please insert the manufacturer of the aethalometer. The authors should clarify why they take into account aerosol absorption at 880 nm.

Authors' response: In this study, we used AE-16 model manufactured from Magee Scientific. In the revised manuscript, we insert the aethalometer model accordingly. Although we got results at two wavelengths (370 nm and 880 nm) from the instrument, manufacturer recommended that results obtained by measuring near-infrared wavelength (880 nm) for analyzing BC concentrations. Data obtained from 370 nm wavelength were usually used to analyze aromatic organic species. Because data at 370 nm wavelength do not have enough sensitivity for analyzing BC concentrations, 370 nm

wavelength of the aethalometer has not been mentioned in the revised manuscrip 💬

We modified text to following text in Page 6 Line 4:

*"The aethalometer (Magee Scientific, AE16) was used to measure the concentration of light absorption particles at 880 nm wavelength."*

**B) Page 7**

Line 3 and following

"Fig.3 depicts monthly variations of the meteorological parameters measured from…

The authors should discuss possible correlations between the considered parameters, and possible variations in these parameters (e.g. temperature trend) during the period considered (2009 – 2015).

Authors' response: we added discussion of variations in temperature trend accordingly. As shown in Figure S1 in this response, the temperature variation does not have meaning due to relatively short period to verify the temperature trend. In addition, trend analysis of temperature and solar radiation are out of the scope of this manuscript. Instead of the trend analysis of temperature and solar radiation, we focused on correlation analysis between solar radiation and CN concentration, which is described in Figure 5 and section 3.2.1.

We added the following sentence at Page 7 Line 25:

*"No clear annual trends of temperature are observed during a six-year period due to a relatively short observation period. In this manuscript, we focused on correlation analysis between temperature (or solar radiation) and CN concentration."* 💬

Line 6 and following

"…the observation site was relatively humid and warm condition compared to other Antarctic stations…"

The statement should be changed to: "..the observation site was relatively humid and warm compared to inland Antarctic stations..".

Authors' response: text was changed accordingly 💬

Line 16:

"(DJF)…..(JJA)"

Should be changed to: …"maximum in the summer (from December to February, DJF) and minimum in the winter (from June to August, JJA)" 💬

Line 20:

"There are no significant anthropogenic sources of aerosol particles in Antarctica, therefore, our results were in good agreement… "

The statement should be changed to: "Our results were in good agreement with the results…".

**C) Page 8**

Line 2 and following:

"The major compounds of aerosol particles found at a coastal Antarctic regions were non-sea salt sulphate and methane sulphonate (MSA) derived from oxidation of DMS produced by phytoplankton (Weller et al., 2011).

Weller et al.'s conclusion (2011) is different. They write referring to Neumayer station: "From thermodenuder experiments we deduced that the portion of volatile (at 125°C) and semi-volatile (at 250°C) particles which could be both associated with biogenic sulphur aerosol, was maximum during austral summer, while during winter non-volatile sea salt particles dominated.".

Atmospheric marine aerosol consists prevalently of primary aerosol (organic material, sea-salt) produced on the ocean surface by bubble bursting and wave crest disruption, and biogenic secondary aerosol (non-sea-salt sulphate and methanesulphonic acid from oxidation of DMS emitted by phytoplankton, and ammonium from biological reduction processes of N-cycle compounds).

During the 2002-2003 summer season, Fattori et al. (2005) reported that the coastal site ("Mario Zucchelli Station" in Terra Nova Bay) was affected by primary and secondary marine input: the sea spray contribution was dominant in the coarse fraction whereas the biogenic source prevailed in the fine fraction.

Authors' response: We understand that the text in the 1$^{st}$ version manuscript was not clear enough, possible to mislead the intention what we wanted to explain. Following referee's suggestion, we changed the sentence as to make the meaning clearer. Weller's conclusion is very well acknowledged, in this manuscript, we mean that highly CN2.5 concentrations during the austral summer season (DJF) most likely to be related to nss-sulphate and MSA derived from oxidation of DMS produced by phytoplankton since secondary formation aerosols play an important role in CN2.5 concentration.

To make clear the intention and meaning, we changed sentence in Page 8 Line 15 as:

"*The high CN$_{2.5}$ concentrations during the austral summer season (DJF) should be related to non-*

*sea-salt sulphate and methanesulphonate (MSA) derived from oxidation of dimethyl sulphide (DMS)*
*produced by phytoplankton (Weller et al., 2011)."*

Line 8 and following

"The CN concentrations typically increase in the summer due to high biological activity, while they decrease in the winter when biological activity is low.…"

The CN concentration is related not only to biological activity, but also to primary aerosol. Primary aerosol includes inorganic salts, inorganic and organic mixture, and biological particles. The contribution of primary aerosol to the total aerosol concentration depends mainly on wind speed and the season, and it is higher in winter and lower in summer. In general, factors affecting total particle number concentrations are the air mass type, meteorological conditions, and whether or not nucleation-mode particles are present.

Authors' response: Our intention was that difference between CN2.5 and CN10 concentrations typically increased in the summer season, whereas those in the winter decreased. Our hypothesis is that the trends of difference should be related to secondary aerosol formation caused by biological activity. Because temperature and solar radiation play an important role in the biological activity, we focused on the correlation between CN2.5 concentration and temperature, and between CN2.5 concentration and solar radiation.

To clarify our intention, we changed text to following sentence in Page 8 Line 21:

*"The difference between $CN_{2.5}$ and $CN_{10}$ concentrations typically increased in the summer season (DJF) due to high biological activity, whereas those in the winter season (JJA) decreased when biological activity is low. Our hypothesis is that trends of the difference should be related to secondary aerosol formation caused by biological activity."*

Line 14 and following

"Our results suggest that CN2.5 concentrations may be more closely coupled with solar radiation intensity than with temperature".

The problem is more complex. An important parameter could interfere, i.e. the ocean temperature, which is different from air temperature. For instance, at the Antarctica research station Aboa Virkkula et al. (2009) observed that the annual maximum daily-averaged particle concentration was later, in February, than the maximum in solar radiation intensity. They concluded that the particle concentrations are more closely linked with the ocean temperature than with solar radiation. Peak sea temperature in polar regions is reached in late summer. As the authors measured both solar radiation

and CN concentration, it could be important to point out if there is a delay between the maximum CN and the maximum solar radiation.

Authors' response: Virkkula el al. (2009) compared daily average particle number concentration and solar radiation from December 2003 to April 2004. Compared with our data-set, it seems that their observation period is not long enough to clearly verify correlation between particle concentrations and solar radiation. Cayan (1980) reported that sea surface temperature and air temperature have roughly same variance. We assume that seasonal trend of ocean temperature should be similar to those of air temperature. We also compare monthly variations of CN2.5, temperature, and solar radiation, as can be seen in Figure S4 of this response. CN2.5 concentration sharply decreased from March, while temperature decrease occurs later, say in May, whereas solar radiation gradually decreased from February. Although temperature gradually decreased in the winter, in addition, CN2.5 concentrations were stable as well as solar radiation. Correlation between CN2.5 concentrations and temperature, and between CN2.5 concentrations and solar radiation would be different month by month. However, relationship among monthly mean values of CN2.5, temperature, and solar radiation was investigated and explained in this study (see Figure 5 and discussions of the manuscript).

[Figure]

Figure S4. A comparison of monthly variations of $CN_{2.5}$ concentrations, temperature, and solar radiation during whole observation period.

In the summer 2014 the CN2.5 and CN10 concentrations are much lower compared to remaining

considered years (Fig.4), but the solar radiation (Fig. 3) remains roughly stable during the summer in the years 2010-2015. These data should be explained.

Authors' response: The reasons for lower CN2.5 concentrations in the summer 2014 could not be explained by solar radiation and temperature because solar radiation and temperature did not show any distinctive variation compared with other years. Other metrological parameters such as wind, air pressure, and RH could not explain the lower CN2.5 concentrations neither. The possible reason is the type of air masses reached to the sampling site. As shown in Figure S5, Case IV where air mass was originated from the South Pacific Ocean was dominant in the summer. Remarkable results in the summer 2014 were that frequency of Case II (air mass was originated from the South Atlantic Ocean) was high and frequency of Case IV was lower than other years. Based on the air mass back trajectory analysis as explained in Sec 2.2 of the manuscript, frequency of four types of air mass were compared in summer season. Seasonal trends of CN2.5 concentrations were different according to air mass history as shown in Figure 13 in the manuscript. In case of Case II, peak CN2.5 concentrations were in November, while maximum CN2.5 concentrations of Case IV were in February. Therefore, it is that increasing frequency of air mass originated from the South Atlantic Ocean (case 2) would explain this lower CN2.5 concentration.

To explain lower CN2.5 concentrations during the 2013-2014 summer season, we added the fowling paragraph on Page 9 Line 7:

"*Unique results of $CN_{2.5}$ concentrations were observed as shown in Fig. 4. The $CN_{2.5}$ concentrations in the summer season of 2013-2014 were much lower than other years. Unfortunately, the reason for the lower $CN_{2.5}$ concentrations could not be explained by solar radiation intensity and temperature because the solar radiation and the temperature did not show any distinctive variation compared with other years. The possible reason is type of air masses reached to the sampling site. Although air mass originated from the South Pacific Ocean (Case IV:* descriptions of the cases I, II, III and IV are described in section 3.3) *was dominant in the summer, based on the air mass back trajectory analysis as explained in Sec 2.2, frequency of air mass originated from the South Atlantic Ocean (Case II) in the summer of 2013-2014 was higher than other years and frequency of air mass originated from Case IV was lower than other years. In case of Case II, peak $CN_{2.5}$ concentrations were in November, while maximum $CN_{2.5}$ concentrations of Case IV were in February. Therefore, it is*

*that increasing frequency of air mass originated from the South Atlantic Ocean would explain this lower CN$_{2.5}$ concentration."*
[Figure]

[Figure]

Figure S5. Frequency of air masses reached to sampling site depending on air mass history in the summer only

Line 17 and following

"The monthly mean CN concentrations increased from September to February…"

Several papers report the annual variation of CN with the maximum in summer and the minimum in winter in coastal areas (Bigg et al., 1984; Gras and Adriaansen, 1985; Gras, 1993; Jaenicke et al, 1992) and in inland stations (Bigg et al., 1984; Samson et al., 1990). A few references should be cited.

Authors' response: We add a few references.

**D) Page 9**

Line 10 and following:

"The clear seasonality of CCN concentrations is probably caused by the seasonal trend of CN concentrations…"

The statement should be changed to: The clear seasonality of CCN concentrations follows the trend of CN.

Authors' response: text was changed accordingly.
[Figure]

**E) Page 10**

Line 3 and following

"The fraction at the SS value of 0.2% during the winter(JJA) was similar to those measured in Mace Head and Finokalia, which are regions representative of marine environment."

"Our observations suggests that the major components in the aerosol particles that are activated to CCN at an SS of 0.2% should be hygroscopic sea salts during winter, while compounds less hygroscopic than sea salt would be dominant during the summer.".

Comparison with the Finokalia site appears inappropriate, as Bougiatioti et al. (2009) performed measurements at Finokalia from mid-June to mid-October (i.e. summer and autumn), not in winter. In addition, Bugiatioti et al. state that "Finokadia is located at a unique "crossroad" of aged aerosol types (marine boundary layer, Saharan desert, European sub-continent, biomass burning events) during the summer period." In the case of King Sejong Station, only a few cases of air masses originated from the continent of South America were shown. The authors' conclusion appears to be oversimplified and should be better explained. In addition to sea-salts, sea-spray aerosol includes organic material (prevalently water insoluble) which possesses a low hygroscopic growth factor, while simultaneously having a CCN activation efficiency higher than soluble non-sea-salt sulphate (Ovadnevaite et al., 2011).

Authors' response: Paramonov et al. (2015) compared results from CCNC measurements 14 sites around world. The sampling sites were just grouped according to location in their study. For instance, Finokalia, Mace Head and RHaMBLe campaign were representative sites for marine environment. However, Bougiatioti et al. (2009) measured physical and chemical characteristics of atmospheric aerosols according to air mass history during the short-term campaign at the Finokalia site. Although they showed different characteristics of aerosols depending to origin and pathway of air masses, it is not clear enough due to results from short-term measurements. To clarify, we have removed the comparison with Finokalia data in the revised manuscript.

In the revised manuscript, text was changed on Page 11 Line 14 as:

*"The fraction at the SS value of 0.2% during the winter (JJA) was similar to those measured in Mace Head, which is a representative site of a marine environment (Paramonov et al., 2015)."*

CCN activation efficiency depends on the constituents of aerosol particles (Dusek et al., 2006; Ovadnevaite et al., 2011). As only CCNC dataset are available in this study, there exists a limitation to infer the chemical compounds of aerosol particles. Thus, we discarded the citation of chemical

compounds (e.g. sea salts) of aerosol particles in the revised manuscript.

We changed sentence to following text in Page 11 Line 17:

*"Although CCN concentrations were low in the winter, our observations suggest that aerosol particles that are activated to CCN during the winter season should be more hygroscopic than those during the summer period."*
[Figure]

Line 8 and following

"Fig. 10 illustrates the seasonal variations in the mean activation ratio of CCN concentrations at an SS of 0.4%..."

As in the previous paragraph, the authors discuss the trend of CCN concentration at SS= 0.2 %, I expect the authors considered the seasonal variations of the ratio between CCN and CN concentrations at supersaturation 0.2%, instead of 0.4%.

Authors' response: The main purpose of this section is to see the seasonal variations of CCN activation ration at SS=0.4%.

To avoid confusion for readers, we divided the section on Page 11 Line 20:

*"3.2.3 Activation ratio and Fitting parameter of CCN."*

**F) Page 11**

Line 16 and following

"The BC concentrations observed at our station were slightly higher than those at other stations in Antarctica…." "Additionally, no clear seasonal patterns were observed in our study throughout the entire observation period.".

The BC concentrations measured by the authors are much higher than those at other Antarctic stations. Please compare the concentrations shown in the paper concerning South Pole, Halley, Neumayer, and Ferraz station (0.65 ng m-3, 1.0 ng m-3, 2.6 ng m-3 and 8.3 ng m-3. respectively), with those measured at King Sejong Station. The very high concentration measured needs to be explained. In addition, it appears to me that a seasonal trend can be noted from Fig.12, i.e. prevalently lower values during winter.

Authors' response: The main scope of this manuscript is to understand seasonal trends of CN and CCN concentrations. Data from CN and CCN concentrations when BC concentrations were higher than 100 ng m-3 were discarded to improve data quality, while we used raw BC data for analysis of trend of BC concentrations. For this reason, mean BC concentrations were slightly high (64.68 ng m-

3). If we discard data when BC concentrations were higher than 100 ng m-3, the mean BC concentrations sharply decreased as 27.43 ng m-3 during whole observation period. However, mean BC concentrations in this study were higher than those measured at other Antarctic stations (e.g. South Pole, Halley, and Neumayer). The reason for the higher BC concentrations might be related to location of sampling site. There are nine permanent on-site stations on the Baton Peninsular of King George Island. In particular, six stations are located within a 10 km radius from the King Sejong Station. There should be extra bias of data from BC measurements due to effect of other stations.

We changed the Figure 12 and modified text of manuscript on Page 13 Line 3:

*"To eliminate effect of local pollution on observations, in this study, data where BC concentrations were higher than 100 ng m$^{-3}$, were discarded. The BC concentrations varied between 1.07 ng m$^{-3}$ and 75.97 ng m$^{-3}$, with a mean of 27.43 ± 4.98 ng m$^{-3}$."*

[Figure]

Figure 12. Monthly mean concentrations of black carbon over the period from March 2009 to February 2015. Here the error bars represents the standard deviation of the measurements from the mean value.

In the revised manuscript, we added fowling text to explain high BC concentration compared with results from other Antarctic station on Page 13 Line 9:

*"The reason of the higher BC concentrations might be related to location of sampling site. There are nine permanent on-site stations on the Baton Peninsula of King George Island. In particular, six*

*stations are located within a 10 km radius from the King Sejong Station.* *There should be extra bias of data from BC concentrations due to effect of other stations.**"*

As already mention of response, it is not main aim of this manuscript to understand BC trend. Data of BC concentrations were used to remove effect of local pollution on CN and CCN analysis. It would seem that there was seasonal trend of BC concentrations during short-term. For example, BC concentrations in the winter were lower than those in the summer during 2011. However, the BC concentrations in the winter were the highest in 2010 and no clear trend of BC concentrations was monitored during 2012. Because the variation of BC concentrations does not have meaning due to relatively short period to verify the BC trend, thus, we do not comment on seasonal trends of BC concentrations in this manuscript.

**G) Page 12**

Line 6

As mentioned earlier in Sec. 2.3

Please change to: As mentioned earlier in Sec. 2.2

Authors' response: Thank you for correction. Text was changed accordingly.

Line 8

"Although they are unreliable due to the low observation frequency…."

I suggest changing this statement to: "The very few cases of air masses originated from the continent of South America show the highest BC and CCN concentrations (Table 1).

Authors' response: text was changed accordingly.

**H) Page 14**

Line 3 and following

"The activation fraction of aerosol particles at the King Sejong Station….was lower than at the Arctic sites indicating that less hygroscopic compounds in aerosol particles should be dominant".

No reference is shown for Arctic sites. The conclusion appears superficial. I recall Ovadnevaite et al.'s paper (2011) which shows that seaspray aerosol enriched in primary organic matter (prevalently hydrophobic) possesses more CCN activation efficiency than more soluble particles dominated by nss-sulphate.

Authors' response: We add a reference in the manuscript. Lathem et al. (2013) showed CCN activation efficiency measured in Arctic area with aircraft during summertime. Based on physical

properties of aerosol particles, it is impossible to exactly understand chemical compounds of aerosol particles. Chemical characteristics of aerosol particles can be just deduced. We modified text to eliminate misunderstanding to readers.

In the revised manuscript, we modified text to following on Page 15 Line 26:
*"It suggests that aerosol particles in Antarctic Peninsula should be less hygroscopic than those in Arctic."*

Line 13
"Although the BC and CCN concentrations were the highest when the air mass originated from the South American continent, the results are not significant because only a small amount of data was analyzed.".
I suggest changing this to: "The very few cases of air masses originated from the continent of South America showed the highest BC and CCN concentrations."
Authors' response: We changed a sentence as referee's suggestion.

*"The very few cases of air masses originated from the South American continent showed the highest BC and CCN concentrations."*

**I) Page 21**
Figure 1 shows devices like OPC and Nephelometer, not used in the measurements.
Authors' response: In this manuscript, we didn't show date form OPC and Nephelometer. We modified Figure 1 to reduce confusion.

[Figure]

CPC: condensation particle counter
SMPS: scanning mobility particle sizer
CCNC: cloud condensation nuclei counter

Figure 1. A schematic diagram for the observation methods used in this study.

References

Baron and Willeke (B&W) Aerosol Measurement, 2nd Edition, J Wiley and Sons, 2001.

Bougiatioti, A., Fountoukis, C., Kalivitis, N., Pandis, S. N., Nenes, A., and Mihalopoulos, N.: Cloud condensation nuclei measurements in the marine boundary layer of the eastern Mediterranean: CCN closure and droplet growth kinetics, Atmos. Chem. Phys., 9, 7053-7066, 2009.

Cayan, D. R.: Large-scale relationships between sea surface temperature and surface air temperature, Mon. Weather Rev., 108, 1293-1301, 1980.

Hinds, Aerosol Technology, 2nd edition, J. Wiley and Sons, 1999.

Paramonov, M., Kerminen, V. M., Gysel, M., Aalto, P. P., Andreae, M. O., Asmi, E., Baltensperger, U., Bougiatioti, A., Brus, D., Frank, G. P., Good, N., Gunthe, S. S., Hao, L., Irwin, M., Jaatinen, A., Jurányi, Z., King, S. M., Kortelainen, A., Kristensson, A., Lihavainen, H., Kulmala, M., Lohmann, U., Martin, S. T., McFiggans, G., Mihalopoulos, N., Nenes, A., O'Dowd, C. D., Ovadnevaite, J., Petäjä, T., Pöschl, U., Roberts, G. C., Rose, D., Svenningsson, B., Swietlicki, E., Weingartner, E., Whitehead, J., Wiedensohler, A., Wittbom, C., and Sierau, B.: A synthesis of cloud condensation nuclei counter (CCNC) measurements within the EUCAARI network, Atmos. Chem. Phys., 15, 12211-12229, 10.5194/acp-15-12211-2015, 2015.

We thank the referee for valuable comments that we have used to improve our manuscript. We have considered the comments and have modified the manuscript accordingly. Our detailed responses to the referee's comments are below.

This manuscript summarizes measurements of aerosol physical properties (size distributions and number concentrations) as well as CCN concentrations performed at a research station in Antarctica. These data are needed to establish quantitative points of reference for a region that may face dramatic changes as a result of climate change. Because of this, this manuscript fulfills an important role and is suitable for publication in ACP.

I do feel that the authors have missed some opportunities in their presentation of the data to dig a little deeper into their data. I would like for the authors to comment on some of the more obvious issues to me:

(1) Very little is mentioned about the particle size distributions. For example, if new particle formation is expected (and there are several references to suggest this in the text), then what do the SMPS data tell us about the nature of the particle formations events. To address this question the data from the size distribution shown in Fig. 8 could show "box-whisker" data that better account for less frequent new particle formation events.

Authors' response: The main purpose is to understand general physical characteristics of aerosol particles at the King Sejong Station in Antarctic Peninsula. Because the study on new particle formation events is out of scope, in the manuscript, brief description about the particle size distributions was done and the particle size distribution data were used to support explanation for monthly trends of CCN concentrations as shown in Fig. 8. In accordance with referee's suggestion, we also showed "box-whisker" data of particle size distribution as shown in Fig. S1. Although there is much outlier in summer season, we don't know that it indicates frequency of new particle formation events. We are preparing other manuscript related to new particle formation events and are

analyzing deeply and carefully about results of the particle size distribution.
[Figure]

[Figure]

Figure S1. Box-Whisker plot of particle size distribution during (a) spring, (b) summer, (c) autumn, and (d) winter season. Red bars in the boxes indicate median values (mean values: green circle), whiskers indicate the 5[th] and 95[th] percentiles, and red cross out of whiskers represents outliers.

(2) Another missed opportunity is the lack of a thorough analysis of the CCNC data. Other than reporting concentrations at specific supersaturations, the "spectrum" of CCN activity is not really discussed in this paper. Some fitting parameters, namely "C" and "k", are presented but one of the most widely used parameters, kappa from k-Koehler theory, is not even mentioned. While I do not demand consideration of these issues as a condition for publication, I do urge the authors to consider how a somewhat minor investment in time might add great value to this publication.

Authors' response: If chemical constituents of aerosol particles are size-dependent, it is complicated to estimate CCN concentrations by using particle size distribution. Because size-dependent chemical information of aerosol particles is not available in this study, an empirical parameterization using in-situ CCN measurements was used in this study. The CCN concentrations as a function of SS were fitted with an equation of the form $Nccn=C(SS)^k$, where Nccn is CCN concentration at a certain SS value, and $C$ and $k$ are the fitting parameters. The fitting was done separately for each SS cycle of CCNC data. Besides that we have focused and found the fitting values c and k values, we tested

single hygroscopicity parameter, kappa in κ-Köhler theory, as suggested by referee. We roughly estimated kappa by using monthly mean CCN concentration at SS of 0.4% and monthly mean total particle concentrations obtained from SMPS data. The kappa is approximately 1.18 in December. The possible reason for the high kappa value would be explained by limitations of measurements available. For example, we decided the critical diameter ($d_{crit}$) by comparing CCN concentration with the integrated SMPS data. In addition, CCN concentrations were measured without size-selection. We decided not to mention the kappa value in the revised manuscript.

The following are a list of edits and concerns, with the page and line numbers listed:

P2, line 5: lager -> larger

Authors' response: text was corrected accordingly.

Line 10: This abstract should not assume that the reader understands what C and k refer to. If you want to include these in the abstract you need to explain what they are.

Authors' response: We agree with referee's opinion. To help reader's understanding, text was added to explain what the $C$ and $k$ are.

We added following sentence on Page 2 Line 9:

*"Based on measured CCN data at each supersaturation ratio (SS),* empirical parameterization were also fitted using formula expressed by power-law function ($N_{ccn}=C\times(SS)^{k}$), where $N_{ccn}$ is the CCN concentrations at a given SS, and $C$ and $k$ are the fitting parameters."

Line 17: improper use of semicolon. Replace with colon

Authors' response: text was corrected accordingly.

P3, line 9: "unascertained" may be a word (I'm not sure) but many simpler words like "undetermined" or "unknown" are better suited.

Authors' response: text was corrected accordingly.

Line 10: remove "their"

Authors' response: text was corrected accordingly.

Line 21: e.g. not i.e. (this is not an exhaustive list of physical properties).

Authors' response: text was corrected accordingly.

P6, line 1: "decided" should be replaced with "determined"

Authors' response: text was corrected accordingly.

Line 2: delete the second "only"

Authors' response: text was corrected accordingly.

Line 7: what model of DMA was used?

Authors' response: We used a cylindrical DMA similar to the TSI 3081 model. The length, inner diameter, and outer diameter of the DMA were 44.42 cm, 0.953 cm and 1.905 cm, respectively.

We added following sentence in Page 6 Line 17:

*"The length, inner diameter, and outer diameter of the DMA were 44.42 cm, 0.953 cm, and 1.905 cm, respectively."*
[Figure]

Line 12: "Besides" might be better replaced with "in addition" (this is clearly a style suggestion)

Authors' response: text was corrected accordingly.

18: "were" should be "was"

Authors' response: text was corrected accordingly.

P7, line 8: Please review this sentence and correct grammar

Authors' response: text was corrected accordingly.

*"The solar radiation varied from 2.3 W m$^{-2}$ to 375.4 W m$^{-2}$, with a mean value of 81.2 ± 38.9 W m$^{-2}$."*

P9, Line 1: Please explain why, for this analysis, a supersaturation of 0.4% was chosen

Authors' response: Anttila et al. (2012) measured cloud droplet number concentration (CDNC) and CCN concentration at five different SS values (0.2, 0.4, 0.6, 0.8, and 1.0%) during the third Palls Cloud Experiment (PaCE-3). The campaign was performed from 11 September to 11 October, 2009. According to their results, they found high correlation between CDNC and CCN concentrations at a supersaturation of 0.4%. The CCN concentrations at less 0.4% supersaturation were lower than CDNC, while the CCN concentrations at higher than 0.4% supersaturation were higher than CDNC.

Based on this result, variation of CCN concentrations at the supersaturation of 0.4% was analyzed in this study.

We added following sentence on Page 10 Line 7:

*"Anttila et al. (2012) measured cloud droplet number concentration (CDNC) and CCN concentrations at five SS values (0.2, 0.4, 0.6, 0.8, and 1.0%) during the third Palls Cloud Experiment (PaCE-3). They showed correlation between CDNC and CCN concentrations at each supersaturation. The relationship between CDNC and CCN concentrations at the SS value of 0.4% was approximately linear, while CCN concentrations were lower than CDNC when the SS value was lower than 0.4% and CCN concentrations at upper 0.4% higher than CDNC. Based on this result, in this study, the supersaturation of 0.4% was chosen to investigate seasonal variations of CCN."*

Line 10: is this redundant with the previous sentence in line 5?

Authors' response: We agree with referee's opinion. Because meaning of two sentences was similar, in same paragraph, we removed the sentence in line 5.

In the manuscript, we remove following sentence in Page 9 Line 5:

*"It was similar to the seasonal cycle of the CN concentrations."*

Line 19: any evidence of growth from Aitken to nucleation mode?

Authors' response: If the referee raises an issue if there is any evidence of growth from Aitken mode particle to CCN size, we cannot provide any direct evidence of this. Nevertheless, we compared monthly mean particle concentration measured from SMPS with CCN concentrations. The calculations showed that the number of accumulation mode particles cannot explain the measured CCN concentrations. It means that the rest of CCN are from the growth of smaller than accumulation mode particles, say from the Aitken mode, or residuals of cloud process. We could not provide direct evidence to distinguish these processes, we modified the sentence as:

Page 11, line1 has been changed to

*"Accumulation mode particles can easily act as CCN (Dusek et al., 2006), hence CCN concentrations increase during the summer and decrease during the winter."*

P10, Line 15: correct spelling of Arctic

Authors' response: Thanks. We checked spelling and changed.

P11, Line 5: how well did this model fit the data? There is no mention of this.

Authors' response: We estimated $C$ and $k$ values by using daily mean CCN concentrations at each SS value. The average correlation coefficient, r, was 0.978. We think it was a good fit.

In the revised manuscript, we added following sentence in Page 12 Line 18:

*"The average correlation coefficient, r, was 0.978."*

Line 17: period (not periods)

Authors' response: text was corrected accordingly.

P13, Line 6: no comma after "for"

Authors' response: text was corrected accordingly.

P14, Line 5: all mention of CCN concentrations need to state the SS

Authors' response: In summary section, we missed SS values for explaining CCN concentrations. It can give readers confusion. Thus, we modified sentence in the manuscript.

To clarify we modified sentence to following text on Page 15 Line 22:

*"In addition, we presented the clear seasonal trends of CCN concentrations at the supersaturation of 0.4%."*

Line 10: again, I don't think the reader knows immediately what C and k refer to. If the authors want this section to summarize results I would suggest explaining this to the reader.

Authors' response: We agree with referee's opinion.

To clarify meaning of $C$ and $k$, we added following sentence on Page 16 Line 3:

*"The C and k are constants were estimated using approximate formula expressed by a power-law function ($N_{CCN}=C \times (SS)^k$) (Twomey 1959)."*

Figure notes: Figs 4, 8, 10-13: what do the error bars represent? This needs to be in the caption.

Authors' response: The error bars represent a standard deviation.

We add caption in Figures 4, 8, 10-13.

*"Here the error bars represent the standard deviation of the measurements from the mean value."*

Figure 9. Why not normalize to total N?

Authors' response: To investigate seasonal variations of fractions of CCN concentration at each SS value in aerosol particles activated CCN at a SS of 1.0%, we normalized it to CCN concentrations at a SS of 1.0% without total N.

Fig 13: seems statistically the same to me.

Authors' response: In this Figure, we found that seasonality of $CN_{2.5}$ concentrations were different in accordance with the air mass history. For instance, the $CN_{2.5}$ concentrations originating from the South Atlantic (Case II) were the highest in November, whereas the $CN_{2.5}$ concentrations originating from the South Pacific (Case IV) were the highest in February as can be seen in Figure 13. This is probably due to difference in chemical compounds that contributed to aerosol formation processes and/or in variations of biogenic activity according to the origin and transport pathway of air masses. This analysis has been explained in the section 3.3. Unfortunately, we don't have chemical data of aerosol particles depending on air mass. To verify our hypothesis, further studies on chemical compositions of aerosol particles need to be carried out in the future.

Fig 14: needs to show something about the variability of the size distributions.

Authors' response: we showed the variation of modal diameter and number concentrations of the size distribution as can be seen in Table 2. In the revised manuscript, the following sentence on Page 14 Line 20 was mentioned.

*"The modal diameters with standard deviation and number concentrations are summarized in Table 2. It is obvious that the modal diameters during the summer are larger than those during the winter for both Aitken and accumulation modes; 0.023 μm in the winter and 0.034 μm in the summer for the Aitken mode and 0.086 μm in the winter and 0.109 μm in the summer for the accumulation mode. The number concentrations for the summer are also higher than the  for the winter for the Aitken and accumulation modes, $49.16 \pm 3.88 \ cm^{-3}$ during the winter and $304.36 \pm 20.10 \ cm^{-3}$ during the summer for the Aitken mode and $44.78 \pm 14.24 \ cm^{-3}$ in the winter and $140.25 \pm 10.64 \ cm^{-3}$ in the summer for the accumulation mode."*

Reference

Anttila, T., Brus, D., Jaatinen, A., Hyvärinen, A. P., Kivekäs, N., Romakkaniemi, S., Komppula, M.,

and Lihavainen, H.: Relationships between particles, cloud condensation nuclei and cloud droplet activation during the third Pallas Cloud Experiment, Atmos. Chem. Phys., 12, 11435-11450, 10.5194/acp-12-11435-2012, 2012.

---

## Author Response (AR2)

[revised manuscript text omitted]

Nevertheless, it is likely that the   increased frequency of air mass  originating from the South Atlantic Ocean (Case II) might have resulted in  the lower $CN_{2.5}$ concentration of the austral summer season of 2013-2014

*These CN concentrations were comparable to the results from the Aboa Station, which is located in the coastal area of Antarctica and is mainly affected by south Atlantic air masses (Virkkula et al., 2009). They showed the daily CN concentrations from December 2003 to January 2007. Although there was variation of the CN concentrations year by year, the daily CN concentrations during astral summer period were ~ 600 
[revised manuscript text omitted]

It is useful to infer hygroscopic properties of aerosol particles with a hygroscopicity parameter, kappa. The kappa values varies from 0 for insoluble particles to larger than 1 for water-soluble salt particles (Petters and Kreidenweis, 2007). The kappa value can be defined by Petters and Kreidenweis (2007) as

$$\kappa = \frac{4A^3}{27 D_{act}{}^3 ln^2 SS}, \qquad A = \frac{4\sigma_w M_w}{RT\rho_w} \qquad (2)$$

where $\sigma w$ is surface tension of water, $Mw$ is the molecular weight of water, $R$ is the universal gas constant, $T$ is the temperature, and $\rho w$ is the density of water. $SS$ is the supersaturation applied in the CCNC. The critical diameter, $Dact$, was estimated following Furutani et al. (2008)

$$\frac{\int_{D_0}^{D_{act}} n(D)dD}{N_{tot}} = 1 - \frac{CCN}{CN} \qquad (3)$$

where *Ntot* is total number concentrations of aerosol particles measured by SMPS. *D* is the electric mobility diameter observed by SMPS. In this calculation, 10 nm was applied for $D_0$ where SMPS scan starts. The CCN/CN ratio indicates the fraction of CCN-active aerosols among total particle concentrations.

In this study, the kappa values were estimated using the monthly mean CCN concentrations at the SS of 0.4%, the monthly mean CN concentrations measured by CPC and the monthly mean size number distribution results obtained from SMPS data. The annual mean kappa value was calculated to be 0.15±0.05. This value is comparable to the previous studies from Artic and subarctic areas. For example, Lathem et al. (2013) who measured the CCN activity at the Arctic by using aircraft measurements reported the kappa value of 0.32±0.21. Martin et al. (2011) inferred total kappa of 0.33±0.13 during cruise observation in Longyearbyen, Svalbard. Kammermann et al. (2010) reported the kappa values varied between 0.07-0.21 in the period of 18 days within the Arctic Circle in Sweden. Jaatinen et al. (2014) also showed the kappa value of 0.13±0.07 using 13-day set of data at subarctic area in Finland (Pallas-Sodankylä station).

**3.2.4 3 Black carbon (BC) concentrations**

[revised manuscript text omitted]

**Anonymous Referee #2**

The authors have addressed most of my concerns. I still have one minor and one major concern.

1) Minor concern: Figure S1, as presented to me in the authors' response, is unreadable. I have not been able to check the supplemental document but if it looks the same the authors need to make it clearer. Just to be perfectly clear, this is what I think a box-whisker plot should look like (I took this random plot from the web):

[Figure]

Authors' response: For the reviewer's and editor's reference, we have redrawn box-whisker plots of particle size distribution as shown in following figures (to show supplementary information of the Figure S1(a)-(d) of the 'author's reply to the reviewers') as:

[Figure]

Figure S1. Box-whisker plot of particle size distribution during (a) Spring, (b) Summer, (C) Autumn, and (d) Winter season.

These new plots used same data-set as the figure 8 of the manuscript, do not show clear increase of recently formed particles. The pattern that the concentration of Aitken mode particles show clear sudden increase during summer season can be seen from the both formats (figure 8 of the manuscript and figure S1 of the reply to the reviewers), authors have chosen to show the seasonal size distribution in the way of mean and standard deviation, as the figure 8 of the manuscript.

2) Major concern: The authors' response to my comment on the CCN measurements does not give me great comfort that the authors have made a careful analysis of their data. While I am happy that the authors took the time to calculate kappa, I would argue two issues:

(a) It is, in fact, quite common to estimate kappa based on bulk measurements of CCN concentrations and SMPS-derived size distributions. As the authors point out, an internal mixture must be assumed. While it's true that this may introduce uncertainties, it is still an important and in fact is an often utilized, method for estimating kappa. Just a quick literature search uncovered three

studies where this was done [1-3], and I am sure there are dozens more.

(b) If the authors feel that they have performed a correct calculation of kappa then they should clearly state their methods and result in their manuscript. It is OK if the value of kappa is unreasonable (and 1.18 is an unreasonable value!!). But it is not OK (in my opinion) to perform a good calculation that provides an unsatisfactory answer and then ignore the result.

Authors' response: We re-calculated the kappa value using monthly mean CCN concentrations at the SS of 0.4%, monthly mean CN concentrations measured by CPC and monthly mean size number distribution results obtained from SMPS data. The critical diameter was estimated using equation introduced by Furutani et al. (2008), as follows:

$$\frac{\int_{D_0}^{D_{act}} n(D)dD}{N_{tot}} = 1 - \frac{CCN}{CN}$$

where $N_{tot}$ is total number concentrations of aerosol particles measured by SMPS. $D$ is the electric mobility diameter observed by SMPS. $D_0$ means smallest size measured by SMPS and $D_{act}$ represents the critical diameter. The *CCN/CN* ratio indicates the fraction of CCN-active aerosols. Subsequently, the kappa value with the critical diameter and SS value can be estimated using the following equation (Petters and Kreidenweis, 2007);

$$\kappa = \frac{4A^3}{27D_{act}^3 ln^2 SS}$$

$$A = \frac{4\sigma_w M_w}{RT\rho_w}$$

where $\sigma_w$ is surface tension of water, $M_w$ is the molecular weight of water, $R$ is the universal gas constant, $T$ is the temperature, and $\rho_w$ is the density of water. *SS* is the supersaturation applied in the CCNC. Re-calculated kappa value was found as 0.15±0.05, a reasonable value compared to the previous studies (Kammermann et al., 2010; Martin et al., 2011; Lathem et al., 2013; Jaatinen et al., 2014).

The possible reason for difference of the kappa value between the previous calculation and the present calculation should be attributed to the difference in the calculation of the critical diameter. In the previous calculation, we decided the critical diameter by comparing CCN concentrations with the

integrated SMPS results (SMPS data integrated from larger to smaller particles: 300 nm to 10 nm). In the previous calculation, the critical diameter was decided when the integrated number concentrations equal to the total CCN concentrations. Because SMPS adopted in this study can scan particle number distribution only up to 300 nm, this might mislead information of the critical diameter in the previous calculation.

In the revised version, we re-calculated the critical diameter and kappa value by comparing fraction of non-CCN-active aerosol particle with the integrated SMPS data (using equation 3 of revised manuscript). The re-calculated kappa value ranged from 0.044 to 0.343, showing an annual mean value of 0.153. These new calculation of the kappa and discussion were added in section 3.2.2.

In the revised manuscript, we added following paragraph for estimation and discussion of hygroscopicity parameter, kappa, on Page 13 Lien 15-Page 14 Line 15:

*"It is useful to infer hygroscopic properties of aerosol particles with a hygroscopicity parameter, kappa. The kappa values varies from 0 for insoluble particles to larger than 1 for water-soluble salt particles (Petters and Kreidenweis, 2007). The kappa value can be defined by Petters and Kreidenweis (2007) as*

$$\kappa = \frac{4A^3}{27D_{act}{}^3 ln^2 SS} \ , \qquad A = \frac{4\sigma_w M_w}{RT\rho_w} \qquad (2)$$

*where $\sigma_w$ is surface tension of water, $M_w$ is the molecular weight of water, R is the universal gas constant, T is the temperature, and $\rho_w$ is the density of water. SS is the supersaturation applied in the CCNC. The critical diameter, $D_{act}$, was estimated following Furutani et al. (2008)*

$$\frac{\int_{D_0}^{D_{act}} n(D)dD}{N_{tot}} = 1 - \frac{CCN}{CN} \qquad (3)$$

*where $N_{tot}$ is total number concentrations of aerosol particles measured by SMPS. D is the electric mobility diameter observed by SMPS. In this calculation, 10 nm was applied for $D_0$ where SMPS scan starts. The CCN/CN ratio indicates the fraction of CCN-active aerosols among total particle concentrations.*
*In this study, the kappa values were estimated using the monthly mean CCN concentrations at*

*the SS of 0.4%, the monthly mean CN concentrations measured by CPC and the monthly mean size number distribution results obtained from SMPS data. The annual mean kappa value was calculated to be 0.15±0.05. This value is comparable to the previous studies from Artic and subarctic areas. For example, Lathem et al. (2013) who measured the CCN activity at the Arctic by using aircraft measurements reported the kappa value of 0.32±0.21. Martin et al. (2011) inferred total kappa of 0.33±0.13 during cruise observation in Longyearbyen, Svalbard. Kammermann et al. (2010) reported the kappa values varied between 0.07-0.21 in the period of 18 days within the Arctic Circle in Sweden. Jaatinen et al. (2014) also showed the kappa value of 0.13±0.07 using 13-day set of data at subarctic area in Finland (Pallas-Sodankylä station)"*

In the revised manuscript, we also added the following sentence on Page 2 Line 17-18 in the abstract section and on Page 18 Line 4-5 in the summary and conclusions section.

*"Furthermore, the annual mean hygroscopicity parameter, kappa, was estimated as 0.15±0.05, for SS of 0.4%."*


This manuscript presents and analyses measurements of aerosol properties (number concentrations, size distributions), cloud condensation nuclei (CCN) and black carbon concentrations for a six-year period (March 2009 to February 2015) at the King Sejong research station in Antarctica.

The Cryosphere and especially Arctic and Antarctica are key components of the Earth's system, and are inherently sensitive to a changing climate serving as the most stunning indicators of climate change. On the other hand, among climate change drivers, aerosols still contribute the largest uncertainty to the total climate forcing estimate especially through the aerosol – cloud interactions. This is due to the great variety of aerosol types, both natural and anthropogenic, their short atmospheric lifetimes and to the subsequent high spatiotemporal variability of their physical and optical properties. The Antarctic continent being the most remote area on the planet from other continents and thus from anthropogenic activities and emissions, it is an ideal place for studying natural aerosol processes in order to understand them and to correctly distinguish between natural and anthropogenic factors influencing the climate. Apart from some long-range transported pollution aerosols, primary aerosol sources like mineral dust, vegetation, soot or secondary aerosols from gas to particle conversion are virtually absent on this almost completely ice-covered continent. Hence, marine air masses advected from the Southern Ocean surrounding the continent, remains the dominant source to the Antarctic aerosol load. Therefore, any dataset of original and accurate measurements that helps to elucidate physical processes taking place in a such climatically sensitive region is important.

In this framework the submitted manuscript is interesting and relevant to the topics of ACP. Moreover it is well written and organised and thus it could be published in the ACP Journal after taking into account the following comments.

The manuscript presents interesting results on Antarctic aerosols based on continuous relatively long term (six years) observations at King Sejong station. The dataset is unique and the analysis of measurements is quite adequate. Core of the manuscript is data analysis on a seasonal basis and at a

next level according to the origin of air masses though timeseries of monthly mean data of some variables are presented (e.g. Fig. 3, Fig. 4, Fig. 7a and Fig. 12). Authors analyse the intra-annual variation of examined parameters and they discuss the main features of their seasonal behavior making an effort to provide possible explanations to interpret their findings. In some cases, they compare their results with that of other research works conducted in Antarctica. However, the whole analysis doesn't go deeper to gain an important insight into the factors determining the aerosol properties seasonality and the factors affecting the CCN activation. For instance, is the new particle formation the only or the main factor that induces seasonal variation in particle concentration (total and CN2.5)?

My main concern is about the gain of the new knowledge that this paper brings. Authors cite in the text, especially in the introduction, several works on Antarctic aerosols and their properties. So, what is the contribution of this manuscript to this knowledge? In the introduction authors state "Although various studies have been performed, the measurements taken at the Antarctic Peninsula and the long-term observations of aerosol particles are still insufficient" but they do not discuss any inter-annual variability or trend (except for the exceptional year of 2014). They focus on the seasonal variation. In order to support their work, authors should clearly state what are the new approach, analysis and/or findings compared to literature and this should be clearly presented in the concluding section as well. Otherwise, they can discuss their results compared to other works, examining whether they are in agreement strengthening thus the existing knowledge since current results are issued from multi-year observations. Actually they do it sometimes. For instance, authors state that the revealed seasonal pattern of CN2.5 and CN10 is in agreement with the results of previous studies (page 8, lines 8-10). I am wondering if the consideration of the seasonal variation of CN2.5 and CN10 separately, is additional and further information compared to previous works. On the other hand, throughout the discussion regarding timeseries, seasonal behavior of CCN concentrations, particles size distribution and CCN activation ratios, there are no references to other relative studies in Antarctica. If this analysis and its findings give new or additional information should be stated by authors adding thus value to their work. The same is valid for the analysis regarding the effects of air masses origin on the aerosols physical properties.

I should however state that the lack of new knowledge doesn't reduce the value of a dataset of original measurements of aerosol properties with a relatively long temporal coverage, in a remote, not easily accessible and very interesting from climate change point of view, area of the planet.

Authors' response:

We appreciate positive feedback from the reviewer. We think the value of this manuscript lies on the fact that although many studies have been performed in Antarctica, research on seasonal variations of CN, CCN, and size number distribution was less conducted in Antarctic Peninsula. In addition, published papers obtained by long-term observations were rare. Based on the multi-year observation in Antarctica, in particular, analysis on characteristics of CCN has been carried out for the first time, to authors' knowledge.

Some minor remarks

• Authors give enough information about instrumentation but they do not discuss any quality control assessment that they apply to their raw records.

Authors' response: To minimize the effect of local source on CN, CCN, and SMPS data, all data were eliminated when wind direction was within 355-55° and the BC concentrations were higher than 100 ng m$^{-3}$ (see Page 5 Line 5-7). When daily and monthly mean concentrations of CN and CCN were estimated with remaining data, subsequently, daily data that the rate of daily data acquisition was higher than 50% were only used to secure the quality of raw data.

In the revised manuscript, we added following sentence on Page 5 Line 7-8.

*"In order to ensure the reliability of measurements, only dataset, of which acquisition rate higher than 50%, were used during all analysis procedures."*

• Authors trying to interpret the exceptional CN concentration levels during the period 2013-2014, found that air masses origin was differentiating this period compared to previous years. Air masses from south Atlantic were more frequent than other years. A comparison of CN concentration levels with analogous measurements (if there are published) at stations which are affected mostly by south Atlantic air masses could support this argument.

Authors' response: Virkkula et al. (2009) showed CN concentrations measured at Aboa Station from December 2003 to January 2007. Aboa Station is located in coastal area in inland Antarctica and is affected by south Atlantic air masses. Although there is variation of the CN concentrations year by year, the daily CN concentrations during astral summer were ~ 600 cm$^{-3}$. It is quite similar value comparing with our results during the period of 2013-2014.

In the revised manuscript, we added the following sentence on Page 10 Line 4-8:

*"These CN concentrations were comparable to the results from the Aboa Station, which is located in the coastal area of Antarctica and is mainly affected by south Atlantic air masses (Virkkula et al., 2009). They showed the daily CN concentrations from December 2003 to January 2007. Although there was variation of the CN concentrations year by year, the daily CN concentrations during astral summer period were ~ 600 cm$^{-3}$."*

• Analyzing the CCN concentration, it was found that its seasonal variation follows the seasonal cycle of particles concentration which is logical. I have however point out that the CCN concentrations during the period 2013-2014 seem to be unaffected by the low particles concentration in that period as they remain similar to other years.

Authors' response:

Authors appreciate the issue raised by the reviewer pointing out the fact that CCN concentration in the summer season of 2013-2014 shows normal value. Unfortunately, we do not have CN10 or SMPS data during the period 2013-2014 (only CN2.5 data are available). Authors think the normal CCN concentrations for 2013-2014 summer season may imply the hypothesis that the nucleation of new particles (2.5nm < Dp < 10nm) was less frequent or weaker for 2013-2014 summer season, judging from the fact that (i) the lower concentration of CN2.5 compared with other summer seasons, and (ii) the fairly normal concentration of CCN (see figure 7 (a)).

For reviewer's reference, in the section 3.2.1, we have been modified to relate frequent air mass origin with the lower CN2.5 concentrations in the summer season of 2013-2014, by adding the following paragraph in revised manuscript on Page 9 Line 26-Page 10 Line 3:

*"Unfortunately, neither CN10 nor SMPS data are available for the austral summer season of 2013-2014 because of mechanical failures, it is not possible to directly explain the low concentrations of CN2.5 for this season in terms of the potential effects of air mass characteristics on the concentration of 2.5-10 nm size particles. Nevertheless, it is likely that the increased frequency of air mass originating from the South Atlantic Ocean (Case II) might have resulted in the lower CN$_{2.5}$ concentration of the austral summer season of 2013-2014."*

• Page 3, lines 8-9. In the sentence "…. the direct and indirect climate effects are still unknown (IPCC, 2013).", I think the word "unknown" is not appropriate. Actually, according to IPCC report, aerosol effects contribute the largest uncertainty in the total radiative forcing. Thus you can replace the word 'unknown' by 'highly uncertain'.

Authors' response: In the revised manuscript, text was changed accordingly, see Page 3 Line 9.

- Page 3, line 21. In the sentence "For these reasons, the observation of the physical properties in Antarctica, …", replace the word 'the' by 'their'

Authors' response: In the revised manuscript, text was changed accordingly, see Page 3 Line 21.

- Page 7, line 14. In the sentence "Fig. 3 depicts monthly variations of the meteorological parameters measured from and automatic weather system (AWS) …" replace the word 'and' by 'an'

Authors' response: In the revised manuscript, text was changed accordingly, see Page 7 Line 22.

- Page 7, line 18. In the sentence "the observation site was relatively humid and warm condition compared to inland Antarctic stations", remove the word 'condition'

Authors' response: In the revised manuscript, we removed "condition" accordingly, see Page 7 Line 26.

- Page 15, line 6. In the sentence "Our results are similar those of previous laboratory and field experiments (Sellegri et al., 2006; Yoon et al., 2007).", add the word 'to' after the word 'similar'.

Authors' response: In the revised manuscript, text was changed accordingly, see Page 17 Line 2.

---

## Author Response (ED2)

Dear Editor:

Thank you for your comments, and please find author's response as below.

Regards,
* * *
1. Page 5: "In order to ensure the reliability of measurements, only dataset, of which acquisition rate higher than 50%, were used during all analysis procedures." Please, clarify this sentence since its meaning is not clear. Probably, replace that sentence with something like "collected, validated data spanned at least 50% of the day". In any case, say more explicitly what has been done.

Authors' response: In revised manuscript, to clarify, we modified the sentence on Page 5 Line 9-10, as:

"To ensure the reliability of measurements, only validated data that spanned at least 50% of one day were used for analysis."

* highlighted in yellow in the 'marked-up' manuscript version

2. Page 13, line 9: "The daily mean kappa values range between 0.07 and 2.19, with a mean of $0.41 \pm 0.10$." What is the utilized symbol here? Obviously, you mean k, not "kappa", right? Please use the correct symbol.

Authors' response: You are right. In the sentence, we mentioned k values not "kappa". Thus, we changed the symbol on Page 12 Line 25- Page 13 Line 1. As:

"The daily mean $k_T$ values  between 0.07 and 2.19, with a mean of $0.41 \pm 0.10$."

* highlighted in yellow in the 'marked-up' manuscript version

3. In any case, be careful to avoid any confusion between the fitting coefficient k (Eq. 1) and the hygroscopicity parameter "kappa". Symbols must be clearly different to each other. I suggest to use another symbol for the fitting coefficient, not "k", in order to avoid confusion.

Authors' response: To avoid confusion between fitting coefficient, k, and hygroscopicity

parameter, kappa, it is good idea to use another symbol for the fitting coefficient. In the revised manuscript, we used another symbol for the fitting coefficient, "$k_T$", and the symbol for the hygroscopicity parameter was not used except Equation (2).

\* highlighted in yellow in the 'marked-up' manuscript version

4. Add one or few sentences in the Introduction, emphasizing what is the novelty of the present study compared to previous ones and what is the new knowledge brought to the readers.

Authors' response: We added the following sentence in the introduction on Page 4 Line 9-13. "Although various studies have been performed in Antarctica, research on the seasonal variations of CN, CCN, and the size distribution of aerosol particles has not been considered in the Antarctic Peninsula. Furthermore, studies based on long-term observations are rare. Multi-year observations involving an analysis of the physical characteristics of CCN are conducted for the first time in this study."

\* highlighted in yellow in the 'marked-up' manuscript version

5. Finally, take care of improving the use of english throughout the text. If possible, let the paper be read and corrected by a native english speaking person.

Authors' response: Following the editor's comment, the revised manuscript has been edited by an English native speaker.

\*\* These grammatical changes are marked as 'RED' in the 'marked-up' manuscript version

[revised manuscript text omitted]

---

## Author Response (ED3)

Dear Editor:

Thank you very much for your valuable comments.

We have applied all your comments in the revised manuscript as attached.

Thank you for all your kind comments.

Best regards,
Young Jun YOON and all co-authors
* * *
Editor comment :

Dear authors,

thank you for your improved manuscript, in which you took into account and addressed my comments. Especially, I do appreciate the efforts to improve the use of english in the manuscript, which has been achieved to a large extent. However, in some instances the applied modifications resulted in change/lost of scientific meaning of relevant sentences. Therefore, I send you the annotated pdf file, which you should be based on and apply the suggested changes on it. Please send me the finalized manuscript which will be acceptable for publication in ACP.

Authors' response:
Authors do appreciate the editor's comment, we applied the suggested changes which are marked as 'RED' in the 'marked-up' manuscript version as below:

[revised manuscript text omitted]

---

## Author Response (AR5)

Dear Editor,

We have applied technical corrections according to editor's comments.

Thank you for all your kind corrections.

Best regards,

Young Jun YOON and all co-authors
* * *
Dear authors,

thank you very much for your corrected manuscript. Only a few technical corrections, according to the annotated pdf file, are now necessary and the manuscript is accepted for publication in ACP.

Best regards

Authors' response:

Authors appreciate the editor's corrections, we applied technical corrections which are marked

as 'RED' in the 'marked-up' manuscript version as below.

We added one more co-author, Ki-Tae Park, who contributed particle concentration analysis.

[revised manuscript text omitted]